# Learning place cells and remapping by decoding the cognitive map

Markus Borud Pettersen[1,2], Vemund Schøyen[2], Anders Malthe-Sørenssen[3], Mikkel E Lepperød[1,3]*

[1]Simula Research Laboratory, Oslo, Norway; [2]Department of Physics, University of Oslo, Oslo, Norway; [3]Department of Biosciences, University of Oslo, Oslo, Norway

## eLife Assessment

This **useful** modeling study shows how spatial representations similar to experiment emerge in a recurrent neural network trained on a navigation task by requiring path integration and decodability, but without relying on grid cells. The network modeling results are **solid**, although the link to experimental data may benefit from further development.

*For correspondence:
mikkel@simula.no

Competing interest: The authors declare that no competing interests exist.

**Abstract** Hippocampal place cells are known for their spatially selective firing and are believed to encode an animal's location while forming part of a cognitive map of space. These cells exhibit marked tuning curves and rate changes when an animal's environment is sufficiently manipulated, in a process known as remapping. Place cells are accompanied by many other spatially tuned cells, such as border cells and grid cells, but how these cells interact during navigation and remapping is unknown. In this work, we build a normative place cell model wherein a neural network is tasked with accurate position reconstruction and path integration. Motivated by the notion of a cognitive map, the network's position is estimated directly from its learned representations. To obtain a position estimate, we propose a non-trainable decoding scheme applied to network output units, inspired by the localized firing patterns of place cells. We find that output units learn place-like spatial representations, while upstream recurrent units become boundary-tuned. When the network is trained to perform the same task in multiple simulated environments, its place-like units learn to remap like biological place cells, displaying global, geometric, and rate remapping. These remapping abilities appear to be supported by rate changes in upstream units. While the model does not learn grid-like units, its place unit centers form clusters organized in a hexagonal lattice in open fields. When we decode the center locations of CA1 place fields in mice, we find preliminary evidence of a similar clustering tendency. This suggests a potential mechanism for the interaction between place cells, border cells, and grid cells. Our model provides a normative framework for learning spatial representations previously reserved for biological place cells, providing new insight into place cell field formation and remapping.

## Introduction

Being able to accurately determine your location in an environment is an essential skill shared by any navigating system, both animal and machine. Hippocampal place cells (*O'Keefe and Dostrovsky, 1971*) are believed to be crucial for this ability in animals. Place cells get their name from their distinct spatial tuning: A single place cell only tends to fire in select locations within a given recording environment (*O'Keefe, 1976*; *Park et al., 2011*).

When an animal is moved between different recording arenas, or a familiar environment is significantly manipulated, place cells can undergo global remapping (*Leutgeb et al., 2004*), wherein spatial

responses are uncorrelated across environments. For less severe changes to the environment (e.g. mild changes in smell or color), place cells can also exhibit less drastic tuning curve changes in the form of partial (*Jeffery, 2011*), rate (*Leutgeb et al., 2005*), and orientation (*Muller and Kubie, 1987*) remapping. Furthermore, geometric modifications of a recording environment elicit distinct place field changes. For example, elongating an environment induces field elongation (*O'Keefe and Burgess, 1996*). Adding a novel wall to a familiar environment may spur so-called field doubling (*Barry et al., 2006*), where a second place field emerges, situated at the same distance from the new wall as the field used to be from the original.

Since the discovery of place cells, a range of other neuron types with navigational behavior correlates have been discovered experimentally. These include head direction cells (*Taube et al., 1990*), grid cells (*Hafting et al., 2005*), border cells (*Lever et al., 2009*; *Solstad et al., 2008*), band cells (*Krupic et al., 2012*), and object vector cells (*Høydal et al., 2019*). Some of these spatial cells can also exhibit changes in their firing profile when an animal is moved between different recording arenas or a familiar environment is sufficiently manipulated (*Fyhn et al., 2007*; *Taube et al., 1990*).

How does the myriad of spatial cell types observed in the brain cooperate to do navigation? One popular theory posits that spatial cells collectively set up cognitive maps of the animal's surroundings (*Tolman, 1948*; *O'Keefe and Nadel, 1978*; *Behrens et al., 2018*). In the past, the term cognitive map has been used colloquially, referring to everything from a neural representation of geometry to charts of social relationships (*Tolman, 1948*; *Tavares et al., 2015*; *Aronov et al., 2017*; *Behrens et al., 2018*; *Whittington et al., 2020*). In this work, we formalize the intuitive notion of a spatial cognitive map by proposing a mathematical definition of it. This serves as a foundation for developing models of spatial cell types and can be used to describe several normative models in the literature.

A range of models has already been proposed in an attempt to explain the striking spatial tuning and remapping behaviors exhibited by place cells. One prominent theory holds that place cell activity results from upstream input from grid cells in the medial entorhinal Cortex (mEC) (*Moser et al., 2008*; *Solstad et al., 2006*; *Jeffery, 2011*). However, there are several experimental findings that challenge this so-called forward theory. For instance, place cells tend to mature prior to grid cells in rodent development (*Langston et al., 2010*; *Wills et al., 2010*). Also, place cell inactivation has been associated with abolished grid cell activity, rather than the other way around (*Morris and Derdikman, 2023*). Another approach is to suggest that non-grid spatial cells are responsible (*Hartley et al., 2000*; *Barry et al., 2006*; *Morris and Derdikman, 2023*). However, the exact origins of place fields and their remapping behavior remain undetermined.

How, then, would one go about modeling place cells in a way that allows for *discovering* how place fields emerge, how remapping occurs, and how different cell types relate? An exciting recent alternative is to leverage normative models of the hippocampus and surrounding regions, using neural networks optimized for a navigation task. When trained, such models learn tuning profiles similar to their biological counterparts (*Cueva and Wei, 2018*; *Banino et al., 2018*; *Sorscher et al., 2023*; *Whittington et al., 2020*; *Xu et al., 2022*; *Dorrell and Latham, 2022*; *Schaeffer et al., 2023*; *Low et al., 2023*). To the best of our knowledge, however, no normative models have tackled the problem of directly learning place cell formation and remapping. Only some address remapping, but do so for other cell types or brain regions (*Whittington et al., 2020*; *Low et al., 2023*; *Schøyen et al., 2023*; *Uria et al., 2020*).

Using our definition of a cognitive map, we, therefore, propose a normative model of spatial navigation with the flexibility required to study place cells and remapping in one framework. In our model, the output representations of a neural network are decoded into a position estimate. Simultaneously, the network is tasked with accurate position reconstruction while path integrating. Crucially, the non-trainable decoding operation is inspired by the localized firing patterns of place cells, but with minimal constraints on their individual tuning profile and their population coding properties.

We find that our model learns representations with spatial tuning profiles similar to those found in the mammalian brain, including place units in the downstream output layer and predominantly border-tuned units in the upstream recurrent layer. We thus find that border representations are the main spatially tuned basis for forming place cell representations, aligning with previous mechanistic theories of place cell formation from border cells (*Barry et al., 2006*).

Interestingly, our model does not learn grid-like representations despite being able to path integrate. Thus, our work raises questions about the necessity of grid cells for path integration. However,

we find that the centers of the learned place fields arrange on a hexagonal lattice in open arenas. This indicates that although grid-like cells are not necessary to form place cells, optimal position decoding still dictates hexagonal symmetry. Inspired by this, we decode center locations for CA1 place fields in mice (data provided by *Lee et al., 2023*), and find preliminary evidence that biological place cells could exhibit clustering in a manner similar to our model.

We train our model in multiple environments and observe that the network learns global, rate, and geometric remapping akin to biological place cells. We find that remapping in the place-like units of the network can be understood as a consequence of sparse input from near-independent sets of upstream, rate-remapping boundary-tuned units. Thus, we show that border cell input can explain not only place field formation, but also remapping.

## Results

### Decoding the cognitive map

The foundation for the proposed place cell model is a learned cognitive map of space. We define a spatial cognitive map as a (vector-valued) function $\hat{\mathbf{u}} \in \mathbb{R}^N$ that minimizes

$$S = \mathbb{E}_t \left[ \mathcal{L}(\mathbf{u}(\mathbf{x}_t), \hat{\mathbf{u}}(\mathbf{z}_t)) + R(\hat{\mathbf{u}}(\mathbf{z}_t)) \right], \tag{1}$$

where $\mathbf{u}(\mathbf{x}_t) \in \mathbb{R}^M$ is some target spatial representation at a true location $\mathbf{x}_t$ (e.g. $\in \mathbb{R}^2$) at a particular time $t$, while $\hat{\mathbf{u}}$ is the learned representation, constrained according to some conditions $R$. Lastly, $\mathbf{z}_t$ is a latent position estimate corresponding to $\mathbf{x}_t$. In our case, we consider a recurrently connected neural network architecture navigating along simulated trajectories. As such, $\mathbf{z}_t$ can be thought of as the network's (internal) position estimate at a particular trajectory step, formed by integrating earlier locations and velocities. For details, refer to Model and objective.

Each entry in $\hat{\mathbf{u}}$ can be viewed as the firing rate of a unit in an ensemble of $N$ simulated neurons. On the other hand, $\mathbf{u}$ is an alternative representation of the space that we wish to represent. In machine learning terms, $\mathcal{L}$ is the loss function, while $R$ is a regularization term. In our case, we want $\mathcal{L}$ to gauge the similarity between the learned and target representations, and for $R$ to impose biological constraints on the learned $\hat{\mathbf{u}}$.

The target representation $\mathbf{u}$ does not need to be of the same dimensionality as $\hat{\mathbf{u}}$, or even particularly biologically plausible. This is evident in several prominent models in the literature (*Dordek et al., 2016*; *Cueva and Wei, 2018*; *Banino et al., 2018*; *Sorscher et al., 2023*) which can be accommodated by the proposed definition in *Equation 1* (see A Taxonomy of Cognitive Maps for complete descriptions). As an example, *Cueva and Wei, 2018* trained a recurrent neural network (RNN) to minimize the mean squared error between a Cartesian coordinate target representation and a predicted coordinate decoded from the neural network (*Cueva and Wei, 2018*). Remarkably, by adding biologically plausible constraints (including 'energy' constraints and noise injection) to this network, the authors found that the learned representations resembled biological (albeit square) grid cells.

As the goal of this work is to arrive at a model of place cells, we will denote the learned representation as $\mathbf{p}$. We take $\mathbf{p}$ to be produced by a neural network with non-negative firing rates, whose architecture is illustrated in *Figure 1a*. Specifically, the network features recurrently connected units (with states $\mathbf{g}$) that project onto output units (with states $\mathbf{p}$), in loose analogy to the connectivity of the Entorhinal Cortex and CA1 subfield of the Hippocampus (*Witter, 2010*; *Leutgeb et al., 2004*).

We constrain the 'energy' of the learned representation by imposing an L1 penalty on its magnitude, use Cartesian coordinates as our target representation, and the mean squared error as our loss. In other words,

$$S = \mathbb{E}_t \left[ |\mathbf{x}_t - \hat{\mathbf{x}}_t|_2^2 + \lambda |\mathbf{g}(\mathbf{z}_t)|_1 \right], \tag{2}$$

where $\mathbf{x}_t$ is a true Cartesian coordinate, $|\cdot|_p$ denotes the $p$-norm, and $\lambda$ is a regularization hyperparameter. Crucially, however, Cartesian coordinates are not predicted directly by the network, but are decoded from the population activity of the output layer. This decoder is non-trainable and inspired by the localized firing profile of place cells. Concretely, we form a position estimate directly from the population activity, according to

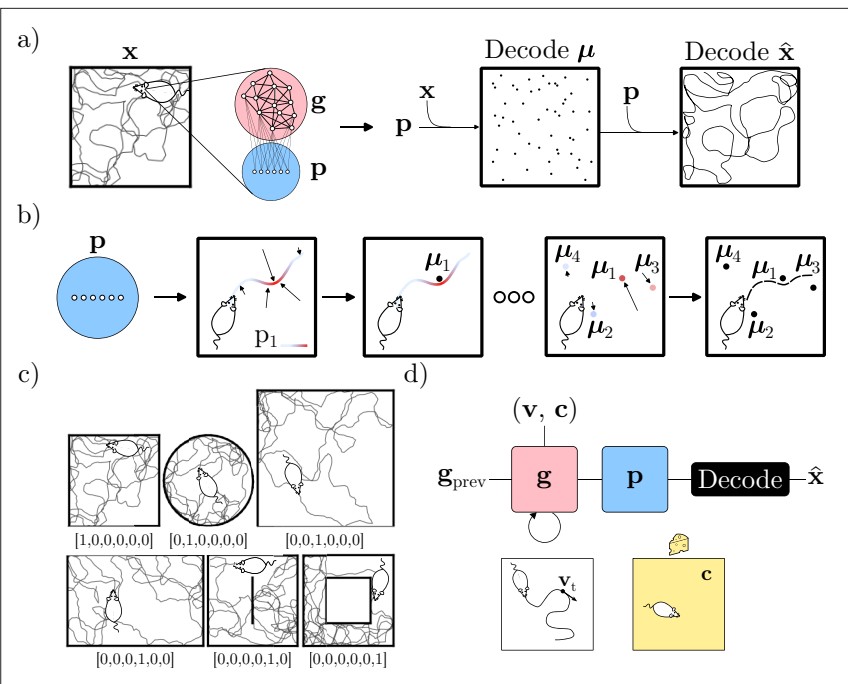

**Figure 1.** The model and task. (**a**) Overview of the decoding approach: Given a simulated trajectory with coordinates **x**, the output states of the network are decoded in terms of their spatial center locations μ, which in turn are used to decode an estimate of the current location $\hat{\mathbf{x}}$. The network is trained to minimize the squared difference between true and decoded positions. (**b**) Illustration of the proposed decoding procedure. For a single unit, the center location is estimated as the average location, weighted by the unit activity along a trajectory. By iterating this procedure, every unit can be assigned a center location. A location can then be estimated as the average center location, weighted by the activity of the corresponding unit at a particular time. Repeating this for every timestep, full trajectories can be reconstructed. (**c**) The investigated geometries, each with an example simulated trajectory. Each environment is labelled by its context signal (one-hot vector). (**d**) Illustration of the network architecture and inputs. **g** features recurrently connected units, while **p** receives densely connected feedforward input from **g**. When moved between environments, the state of the RNN is maintained ($\mathbf{g}_{\text{prev}}$). The input **v** denotes Cartesian velocities along simulated trajectories, while **c** is a constant (in time and space) context signal.

$$\hat{\mathbf{x}}_t = \frac{\sum_{i=1}^{N} p_i(\mathbf{z}_t)\boldsymbol{\mu}_i}{\max\left(\varepsilon, \sum_{i=1}^{N} p_i(\mathbf{z}_t)\right)}, \tag{3}$$

where, again, $N$ is the number of output units and $\varepsilon$ is a small constant to prevent zero-division, while

$$\boldsymbol{\mu}_i = \frac{\mathbb{E}_t[p_i(\mathbf{z}_t)\mathbf{x}_t]}{\max\left(\varepsilon, \mathbb{E}_t[p_i(\mathbf{z}_t)]\right)}, \quad i = 1, 2, 3, ..., N \tag{4}$$

is the estimated *center* of a given output unit. Note that the decoding essentially just consists of two soft maximum operations: *Equation 4* estimates the location of a cell's maximal activity and *Equation 3* yields a predicted position using a weighted average (i.e. an approximate center of mass) of unit activity and their corresponding center locations.

Intuitively, if one cell is highly active at a particular location, its center location will be pulled toward that position. If the centers of the entire ensemble can be established, a position estimate can then be formed as a weighted (by firing rate) sum of the ensemble activity. If multiple units in a particular region of space are co-active, the position estimate is pulled towards the (weighted) average of their center locations. This approach allows us to extract a position estimate directly from the neural ensemble without knowing the shape or firing characteristics of a given unit. This should encourage minimally constrained, yet place-like representations.

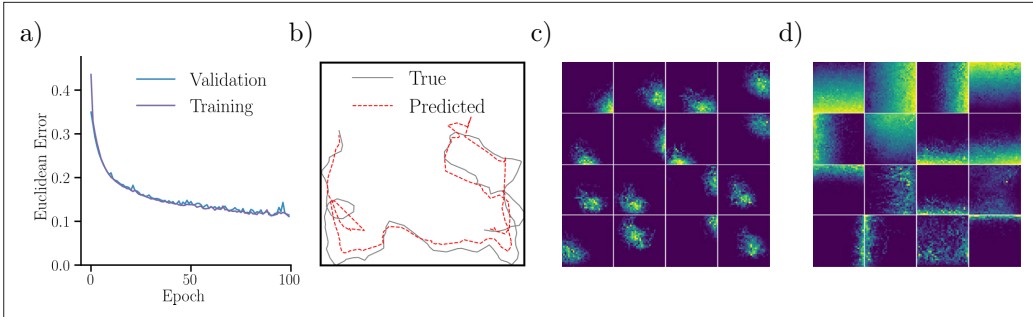

**Figure 2.** Trained network performance and representations. (**a**) Euclidean distance (error) between target and reconstructed trajectories over training time. Shown is the error for both training and validation datasets. (**b**) A slice (timesteps 250–400) of a decoded trajectory (dashed, red) and the corresponding target trajectory (black). (**c**) Ratemaps for the 16 output units with the largest mean activity, in the square environment. (**d**) Same as (**c**), but for recurrent units.

*Figure 1a* provides a high-level overview of the proposed decoding scheme and the network explored in this work, and *Figure 1b* provides a more detailed account of how output unit activity is decoded to estimate the network's position.

The network is tasked with minimizing *Equation 2* while simultaneously path integrating (see 4.1 for details) along simulated trajectories in six distinct environments. Each environment, along with example trajectories is shown in *Figure 1c*. To discriminate between environments, the network is also provided with a constant context signal that uniquely identifies the geometry. An overview of the network architecture and inputs is given in *Figure 1d*, and each context signal is inset in *Figure 1c*. Notably, our network is not given any positional information and must infer its position by traversing and learning the geometry of the environment.

## Learned representations and remapping

With a model in place, we proceed by investigating the learned representations and behaviors of the trained network. *Figure 2a* shows the evolution of the decoding error (the average Euclidean distance between true and predicted trajectories) as a function of training time for the RNN. The validation set error closely trails the training error, and appears to converge around 0.15. The error is computed as an average over all six environments and over full trajectories (time). Thus, the decoding error includes initial timesteps, where the network has no positional information. Disentangled errors for each environment and along trajectories (time) can be seen in supplementary Fig. *Appendix 1—figure 1*, showing how different environments have different error profiles, and how errors decrease after some initial exploration. This can also be seen in *Figure 2b*, which showcases a slice of a true and corresponding predicted trajectory for timesteps 250–400 in the square environment.

Having established that network predictions align with target trajectories (confirming that the network has learned to path integrate), we consider the learned representations of the network. *Figure 2c* displays ratemaps of the 16 most active output units in the square environment. Notably, the responses of these units resemble biological place cell tuning curves. The learned place fields appear unimodal, isotropic, and with a monotonically decaying firing rate from their center, much like a Gaussian profile. However, some units display more irregular fields, especially near boundaries and corners. Responses of the most active recurrent units resemble biological border cells (*Figure 2d*). For both the output and recurrent layers, a large fraction of units are silent or display no clear spatial tuning (see *Appendix 1—figures 2 and 3* for ratemaps in all environments). For example, in the square arena, approximately half of all output units are silent.

Interestingly, when comparing network spatial responses across environments, units display changes in their tuning curve. This effect can be clearly observed in unit ratemaps shown in *Figure 3a*. In the numerical experiment, the trained network is first run in the square environment (context A), before being moved to the square with a central wall (context B), and subsequently returned to the original square (context A'). Visually, many output units exhibit marked shifts in their preferred firing location when the network is moved between contexts (i.e. transfers A to B or B to A'). However, returning to the original context appears to cause fields to revert to their original preferred firing

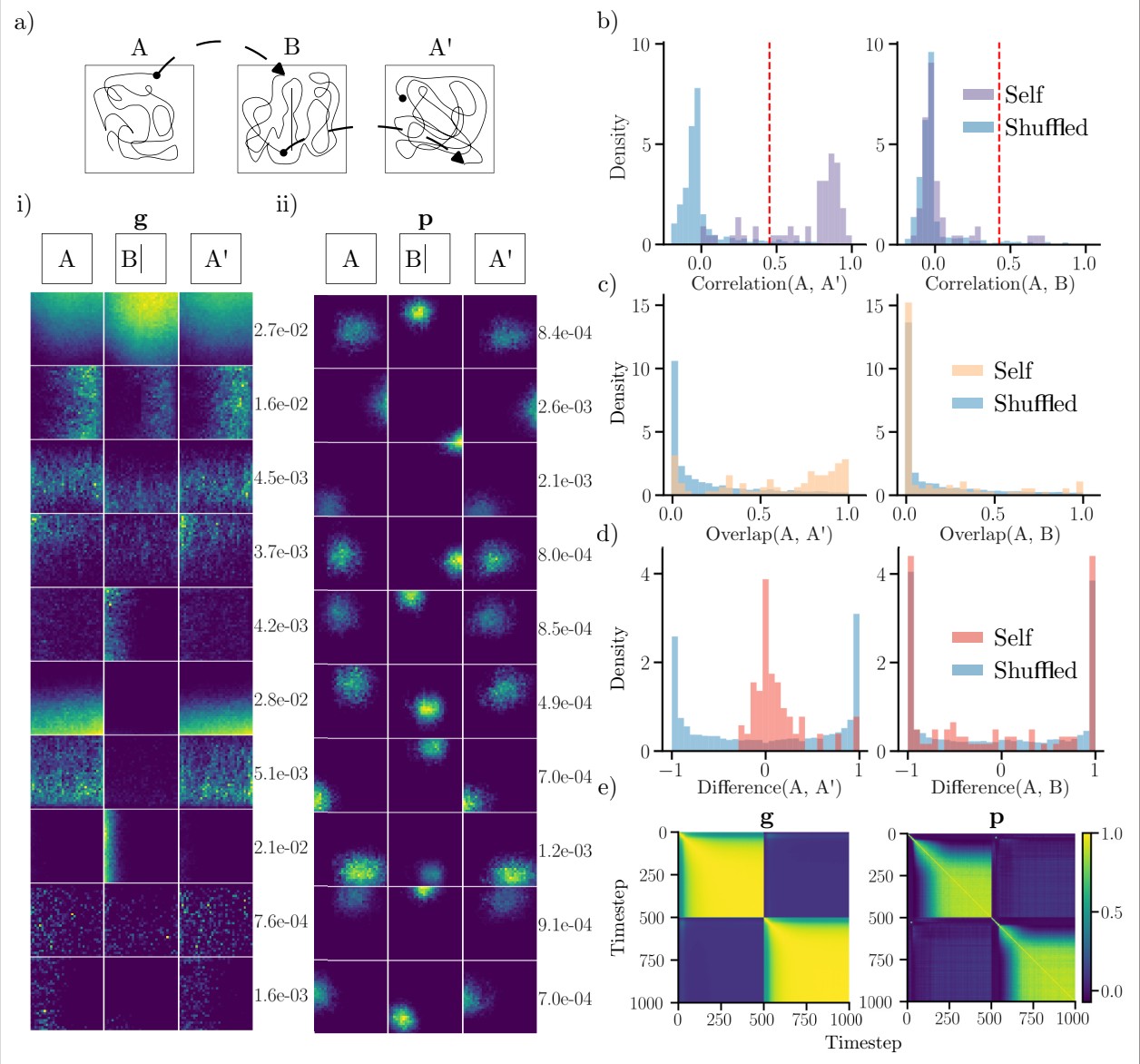

**Figure 3.** Comparing representations across environments. (**a**) Top: The network is run in a familiar square environment (**A**), transferred to the square with a central wall (**B**), and revisits the original square (**A'**). The network state persists between environments, and starting locations are randomly sampled. Bottom: (i) Ratemaps for a subset of recurrent units (**g**) with the largest minimum mean rate across arenas. Rows represent unit activity, with max rate inset on the right. (ii) Same as (i), for output units. (**b**) Distribution of spatial correlations comparing ratemaps from active units across similar contexts (**A, A'**) and distinct contexts (**A, B**). Shuffled distributions are formed by randomly pairing units across contexts. The dashed red line indicates the 95th percentile of the shuffled distribution. (**c**) Distribution of rate overlaps for all units with non-zero activity in any environment. (**d**) Distribution of rate differences. (**e**) Ratemap population vector correlations for units with non-zero activity at every timestep for transitions (timestep 500) from A to B.

locations. In addition to firing location modifications, units also exhibit distinct rate changes. Besides output units, recurrently connected units also display remapping behavior when the network is moved between environments. As shown in the unit ratemaps of *Figure 3a* and (i), boundary units exhibit rate changes. In particular, several units are silenced when moving between conditions. However, none of the included units exhibit changes in their preferred firing location. Thus, recurrent units appear to remap mainly through pronounced rate changes, which we also demonstrate in Recurrent units rate remap.

That the network exhibits remapping-type behaviors is supported by multiple analyses (*Figure 3b–e*). In particular, the distribution of output unit spatial correlations across different environments (A and B) matches that expected from a shuffled distribution. Conversely, correlations

comparing different visits of the same environment (A and A') are different from a shuffled distribution (*Figure 3b*). This behavior is consistent with global remapping behavior (*Leutgeb et al., 2004*; *Leutgeb et al., 2005*). Notably, the network's remapping occurs with fixed weights (i.e. after training). Rate overlaps (*Figure 3c*) display similar distributional properties: Comparing across environments yields rate overlaps resembling those from a shuffled distribution, and comparing similar environments yields higher rate overlaps.

Rate differences (*Figure 3d*) also follow the same trend. In this case, the difference in rates between A and A' is chiefly zero-centered and approximately symmetric, suggesting that there are only small rate changes when revisiting an environment. The rate difference between environments (and between shuffled units), is also roughly symmetric. However, in this case, the distribution is bimodal with peaks corresponding to maximally different rates. Thus, a large number of output units are active in only one environment. Again, the distribution of differences between distinct contexts closely trails a shuffled distribution. As shown in *Figure 3e*, ratemap population vector correlations mirror the transition between environments, both for recurrent units and output units. Included are correlations for the transition from A to B (at timestep 500). Notably, there is a sharp drop-off in correlation at the transfer time, demonstrating that population vectors are uncorrelated between environments for both unit types. Conversely, ratemaps are highly correlated within an environment. However, there is a time delay before maximum correlation is achieved.

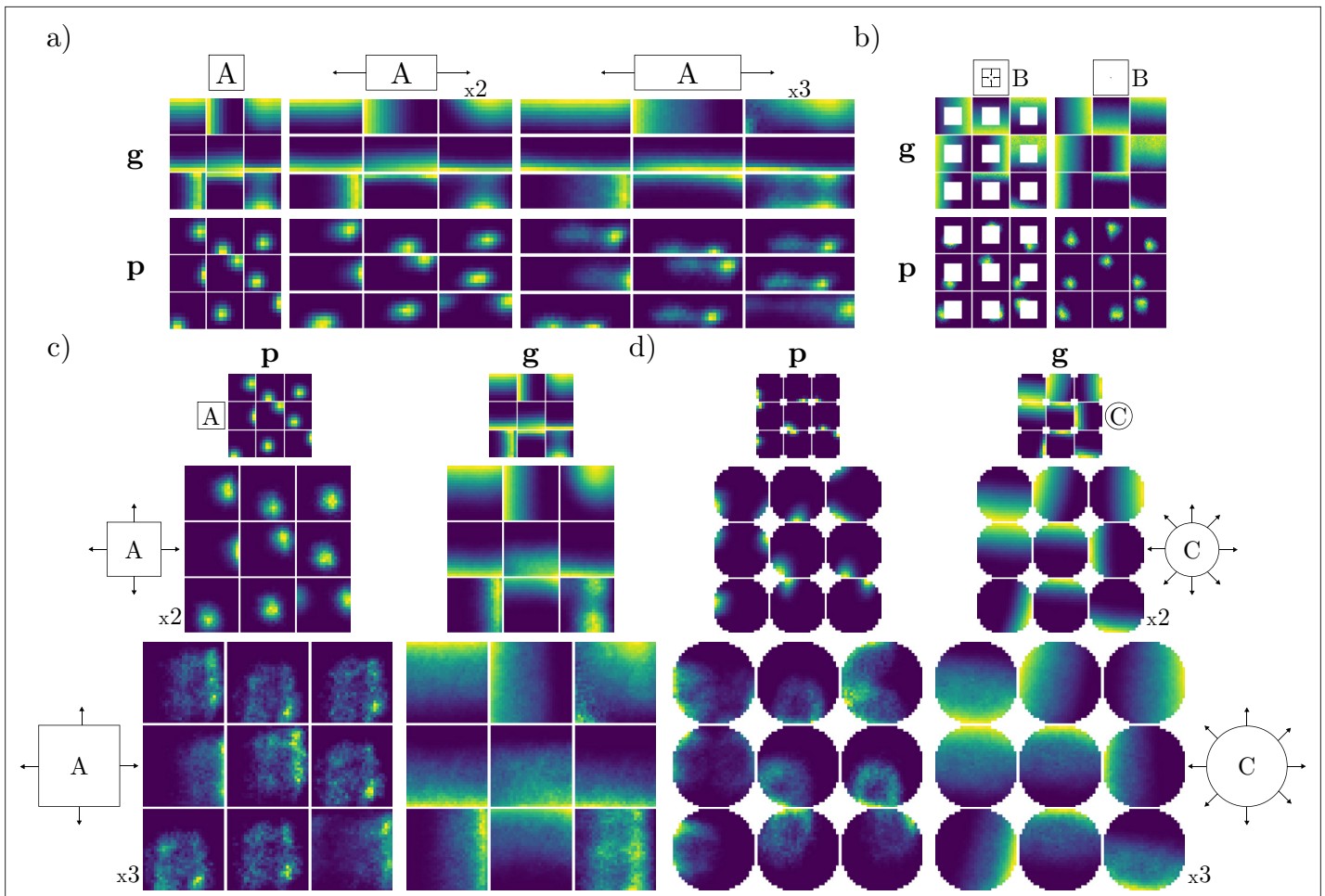

**Figure 4.** Effects of geometric manipulations on learned representations while maintaining the original context signal. (**a**) Ratemaps of ine recurrent (**g**) and output units (**p**) during horizontal elongation of a familiar square context. The top inset indicates the geometry and context signal (**A**), as well as manipulation of the environment (horizontal stretch by factors of 2 and 3). (**b**) Similar to (**a**), but the geometric manipulation consists of filling in the central hole of the familiar context (square with central hole, context B). (**c**) Similar to (**a**), but for joint horizontal and vertical elongation. (**d**) Similar to (**c**), but for uniform expansion of a familiar circular environment (**C**).

Together, these findings demonstrate that the model learns place- and border-like spatial representations. Moreover, output units exhibit global remapping between contexts, whereas recurrent units mainly rate remap.

## Effects of geometric manipulations

In addition to remapping between different contexts, we show that manipulating familiar geometries induces distinct representational changes. In particular, *Figure 4a* shows how unit ratemaps respond as the familiar square environment is elongated horizontally. Intriguingly, the learned place-like fields of the output units appear to expand with the arena. For sufficient elongation, responses even appear to split, with an additional, albeit weaker firing field emerging (e.g. lower right output unit, 2 x elongation). Elongation behavior has also been observed in biological place cells in similar experiments (*O'Keefe and Burgess, 1996*).

Expanding the square also elicits a distinct response in recurrent units: Unit firing fields extend to the newly accessible region, while maintaining their affinity for boundaries. A similar effect can be observed in *Figure 4b*, where the central hole is removed from a familiar geometry. In this case, both recurrent and output units perform field completion by extending existing firing fields to previously unseen regions. This shows that the network is capable of generalizing to never-before-seen regions of space.

Finally, we also considered the effects of expanding environments in a symmetric fashion. Included are results for the familiar square (*Figure 4c*) and circular (*Figure 4d*) environments. Unlike the single-axis expansion in *Figure 4a*, network representations expand symmetrically in response to uniform

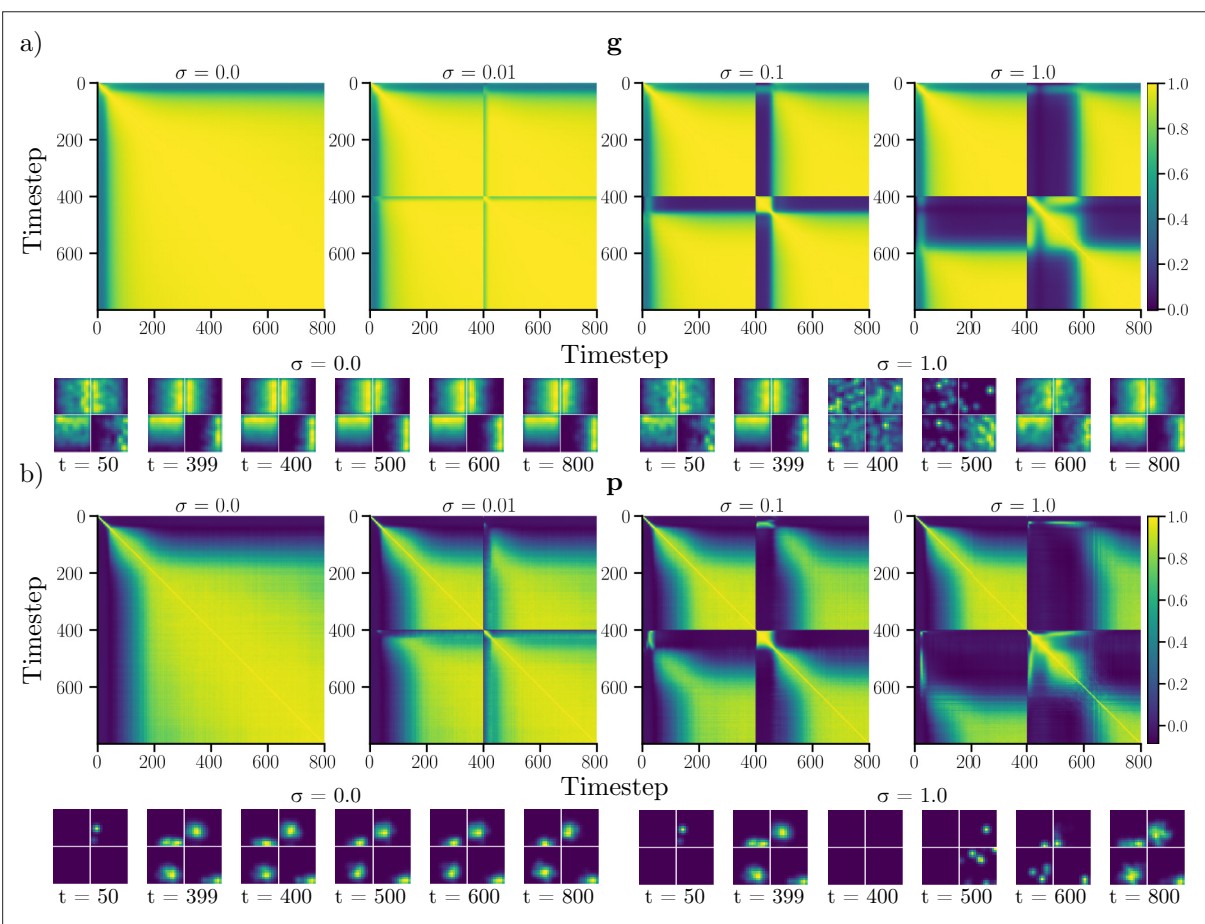

**Figure 5.** Effects of noise injection during navigation. (**a**) Ratemap population vector Pearson correlation between timepoints of 800-step trajectories in the square environment. At timestep 400, additive Gaussian noise (with standard deviation $\sigma$) is injected into the recurrent state (**g**). The top row shows correlations for different noise levels ($\sigma = 0, 0.01, 0.1$, and 1.0). The bottom row features ratemaps of the four units with the largest mean activity, at different timepoints. Ratemaps are shown for $\sigma = 0$ and $\sigma = 1.0$. (**b**) Same as a), but for output units (**p**).

expansion. However, some output units display distinct field doubling (see e.g. 2x expansion; middle inset for both *Figure 4c*, bottom right, and *Figure 4d*, middle row). For large expansions (3 x), output responses become more irregular. However, in the square environment, there are still visible subpeaks within unit ratemaps and some output units reflect their main boundary input (with greater activity near one boundary). Recurrent units, on the other hand, largely maintain their firing profile. In the circular environment, some output units display an almost center surround-like profile (e.g. 3 x expansion, middle row, two rightmost units). This peculiar tuning pattern is an experimentally testable prediction of our model.

## Representations are attractive

We have demonstrated that the RNN exhibits signs of global remapping between different familiar contexts, and field changes when a familiar geometry is altered. In this section, we further explore the behavior of the network when perturbing its internal states out of its normal operating range. Finally, we also discovered possible mechanisms supporting the network's remapping ability.

The first analysis consists of injecting noise into the recurrent state of the network, to determine whether it exhibits attractor-like behavior. *Figure 5* shows the resulting ratemap population vector correlations for an 800-step trajectories in the square context, when noise is injected at the midpoint of the sequence. When no noise is injected ($\sigma = 0$), both recurrent units (*Figure 5a*) and output units (*Figure 5b*) quickly settle into a stable highly correlated state. Unit ratemaps reveal that this state corresponds to network units firing at their preferred locations.

When noise is injected, ratemap correlations temporarily decrease, before the network settles back into a steady-state configuration. Importantly, states before, and long after noise injection are highly correlated. We observe that this is also the case for moderate amounts of injected noise, as can be seen from unit ratemaps for $\sigma = 1.0$. We also observe that the time required to reach a steady state increases in proportion to the amount of noise injected. Thus, even though the network was trained without noise, it appears robust to moderate perturbations. This suggests that the learned solutions form an approximate attractor. To verify that this attractive behavior is not solely due to error correction following boundary interactions, we also conducted a similar experiment, in which velocities are ablated at noise injection (see Velocity Ablation).

Beyond the fact that the network appears to converge towards particular, steady-state representations after noise injection, we also note that each environment appears to correspond to distinct attractive states (as evidenced by the global-type remapping behavior). To uncover why the network converges to a given representation, we conducted a simple context mismatch experiment (see Representations are guided by context for details and ratemaps), which suggests that the context signal determines the resulting representation, up to geometric deformations (as in *Figure 4*).

To further explore the network's behavior, we applied dimensionality reduction techniques to network states along a single trajectory visiting all geometries (and contexts). Remarkably, we find that a low-dimensional projection of the recurrent state captures the shape of each traversed environment. The top row of *Figure 6a* showcases a 3D projection of the recurrent state, where each point is color-coded by the visited environment. Besides reflecting the shape of the environment, the low-dimensional projection also showcases transitions between environments. For output units (bottom row of *Figure 6a*), the low-dimensional projection consists of intersecting hyperplanes that appear to maintain some of the structure of the original geometry. For example, states produced in the square with a central hole, appears to maintain a central void in the low-dimensional projection. The difference between recurrent and output states may reflect the pronounced sparsity of the recurrent layer, as well as the observed reuse of output units during remapping. In other words, a large number of recurrent units are mutually silent across environments, which could make for easily separable states. In contrast, a larger fraction of output units are used, and reused, across environments, leading to entangled and less separable states.

Using PCA, we find that the recurrent states of the network within an environment can be well described using just a few principal components (four principal components explains >90% of the variance). For reference, *Figure 6b* showcases the fraction of explained variance, as well as the cumulative variance of the recurrent state. However, the same is not true for the full trajectory visiting all environments (requiring around 20 principal components to achieve a similar amount of explained

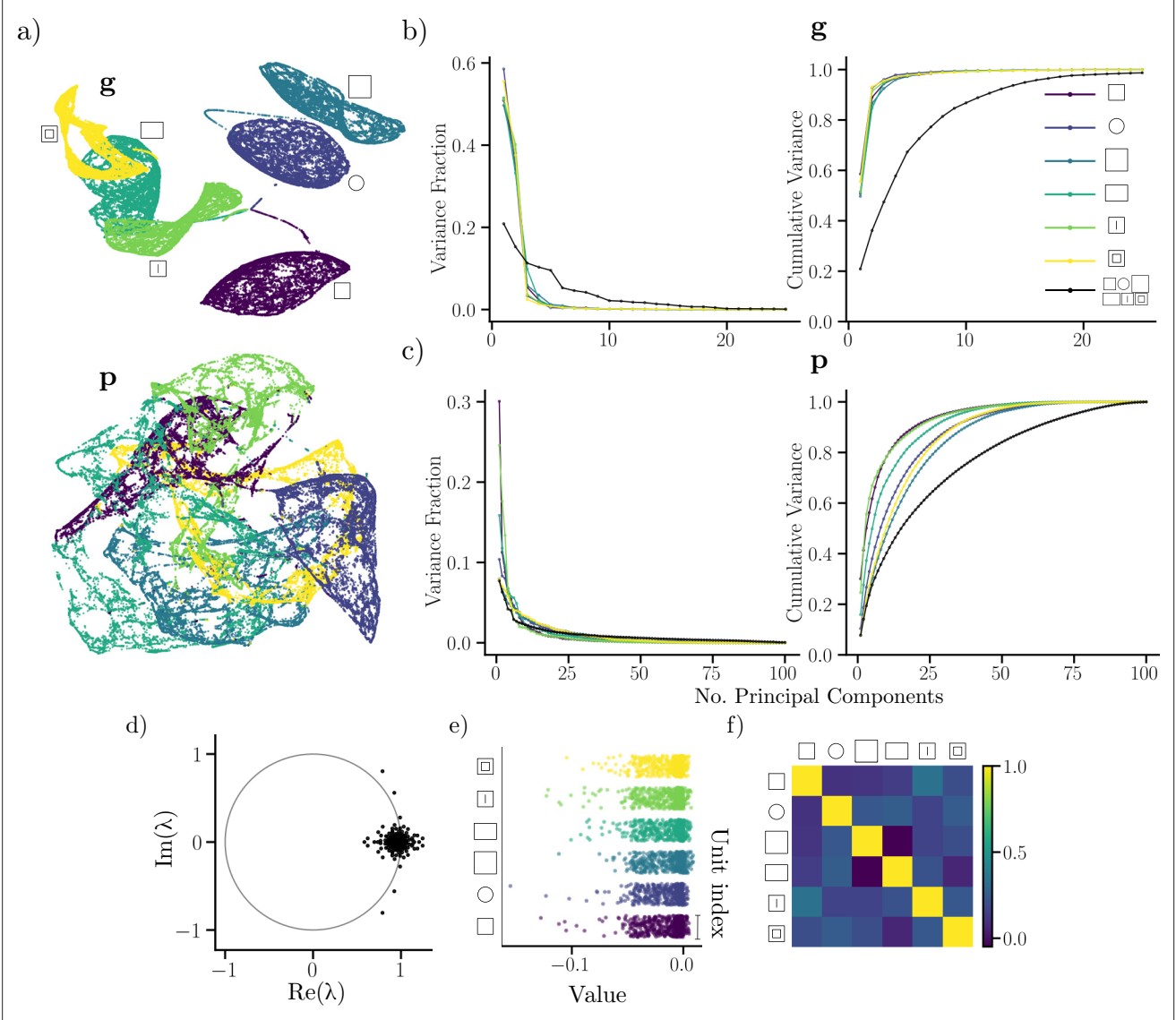

**Figure 6.** Low-dimensional behavior of the trained recurrent network. (**a**) Low-dimensional UMAP projection of the recurrent (top) and output unit (bottom) activity for a trajectory visiting all six environments. The color of a point in the cloud corresponds to the environment identity. (**b**) Fractional and cumulative explained variance using PCA for recurrent units for each environment. (**c**) similar to (**b**) but for output units. (color scheme as in **a**). (**d**) Eigenvalue spectrum of the recurrent weight matrix. The unit circle (gray) is inset for reference. (**e**) Jitter plots of context weights corresponding to each environment. For every environment, the weight to each recurrent unit is indicated. (**f**) Pearson correlation between context weights corresponding to different environments.

variance). This hints that the multi-environment representation can be factorized into several independent, low-dimensional representations, possibly one for each environment.

A similar trend is evident for output unit responses (shown in *Figure 6c*). However, in this case, a larger number of units is needed to explain a substantial fraction of the state variance for each environment (>25 for approximately 70–90% explained variance) with noticeable differences between environments. Also, almost all (>75, out of 100) principal components are required to account for the full output state across environments. It thus appears that more independent units are active within a given environment, and that all 100 units are involved in encoding the full set of environments.

To begin exploring possible mechanisms supporting remapping, and the apparent independence of network states across environments, we investigated the weights of the recurrent layer. *Figure 6d* shows the eigenvalues of the recurrent weight matrix. It has several eigenvalues with above-unit magnitude. In other words, the RNN is potentially unstable. However, as shown in *Appendix 1—figure 1*,

the network exhibits stable decoding errors, even for very long sequences. Moreover, we know from *Figure 5* that the network is stable in the face of transient noise injection. One possibility is that the nonlinearity of the recurrent network ensures stability. Another interesting possibility is that large eigenvalues are associated with remapping, where unstable eigenvectors correspond to transitions between discrete attractors representing distinct environments.

We have already demonstrated that the context signal apparently fixes the global representation (see Representations are guided by context) but the question still remains *how* the network switches between representations. While a complete description depends on the non-linear behavior of the full RNN, we observe a relationship within the context input weights that could shed light on the behavior in the network. Concretely, we find that a large proportion of context weights are negative (*Figure 6e*), and the rows of this matrix are largely uncorrelated (*Figure 6f*). Thus, the context signal (which is non-negative) could inhibit independent sets of units, leading to sparse and orthogonal recurrent units across environments through rate changes, and providing a simple mechanistic interpretation of the observed remapping behavior.

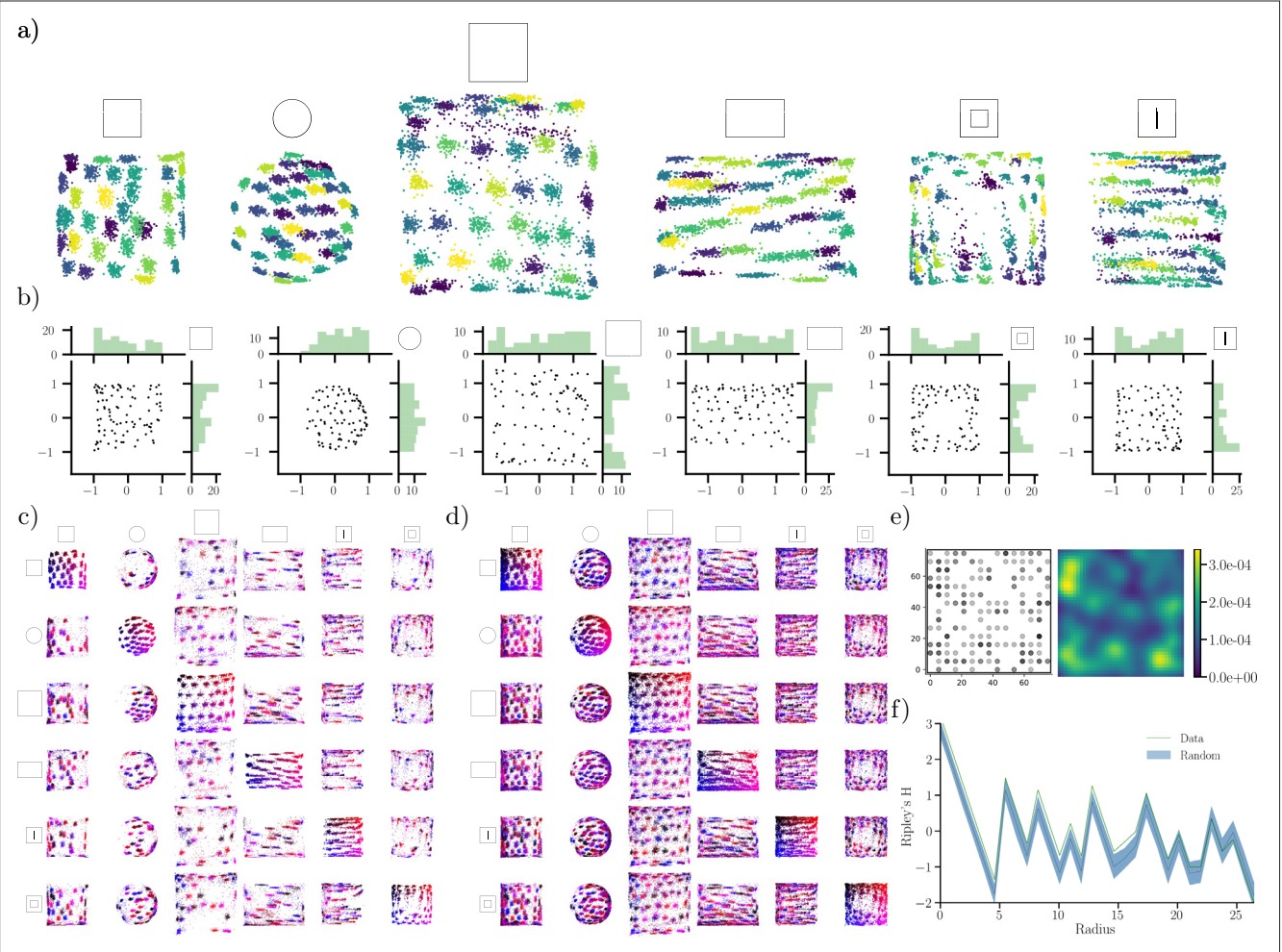

**Figure 7.** Output unit centers reside in hexagon-like arrangements. (**a**) Center locations for each output unit (**p**) in every geometry, decoded from 100, 30000-timestep trajectories for units with high spatial information. Decoded centers are shaded by unit identity. (**b**) Center locations and marginal distributions of centers in each environment, for active units along a single trajectory. (**c**) Displacement of centers between environments for units with high spatial information. Every unit is color-coded by its spatial location in the environment on the diagonal. For each row, the distribution of the included units are shown in every other environment. (**d**) Same as (**c**), but for all units. (**e**) Experimental CA1 place field centers decoded from ratemaps for a mouse foraging in a square 75×75 cm environment (left) and corresponding kernel density estimate (right). (**f**) Ripley's H for the field centers in (**e**) and random (uniform) distributions on the same 15×15 grid as in **e**. The shaded region indicates two standard deviations for 100 random, uniform samplings of the grid.

## Distribution of learned centers

Experimentally, place fields appear to be irregularly distributed throughout large environments with a small increase in the number of fields near boundaries (*Harland et al., 2021*). Place field phases have also been shown to correlate with the peak firing locations of grid cells (*Whittington et al., 2020*). We, therefore, explore whether there is structure to the spatial arrangement of the model's learned place fields.

*Figure 7a* shows the arrangement of decoded centers for all units, collected over 100 long-sequence trajectories, in each environment. In other words, for each cell in the population, their centers are decoded 100 times, one for each trajectory. Surprisingly, we find that the decoded centers tend to reside on the vertices of a semi-hexagonal grid, especially in larger symmetrical geometries. This effect is especially evident in the square and large square environments. However, in all environments, this grid structure exhibits distortions, and in the case of an anisotropic environment (the rectangle), the grid is clearly elongated along the horizontal axis. Our findings accord with the notion that place fields are likely to reside on the vertices of a hexagonal grid (*Whittington et al., 2020*). These findings also resonate with the observations that grid cell patterns deform in non-symmetric environments and novel environments (*Krupic et al., 2015*; *Barry et al., 2012*; *Ginosar et al., 2023*). However, our model does not feature any grid-like units. Together with the fact that our model is optimized for decoding, it appears likely that such an arrangement is somehow advantageous for encoding and decoding location.

In *Figure 7b* we display an example decoding of centers along with their one-dimensional marginal distributions. We find that the centers seem to cluster somewhat along the borders of the environment, similar to experimental observations in *Harland et al., 2021*, which might reflect the upstream boundary input of our model's output units. Unlike the aggregate over multiple trajectories, the single-trajectory decoding does not reveal an equally pronounced hexagonal arrangement. Besides exhibiting a striking hexagonal distribution within an environment, we also observe that there is no apparent pattern to the transformation of center locations between environments. This once again supports the finding that units undergo global-type remapping between environments (see *Figure 7c–d*) where color coding is relative to position in the environment along the diagonal.

To investigate whether fields in biological place cells display center clustering similar to our model, we decoded the field centers of CA1 place cells (data provided by *Lee et al., 2023*), see Ripley's H & Clustering and Experimental Phase Distributions for details and extended results. *Figure 7e* shows a distribution of place field centers from one example animal (QLAK-CA1-50), and a corresponding kernel density estimate. We can see that field centers cluster near boundaries. Moreover, there appears to be a tendency for the clusters to arrange in a hexagonal fashion, similar to our computational findings.

To further quantify the regularity in the spatial arrangement of field centers, we considered Ripley's H function, which gauges the average number of points falling within a particular radius of any other point relative to a uniform baseline, thus providing a measure of clustering in point data. *Figure 7f* shows Ripley's H for the field centers, as well as a random baseline sampled on a 15×15 grid matching the experimental ratemap resolution. We find that Ripley's H is larger for the experimental data than for random, uniform samples at small and intermediate scales. This indicates that the place field centers cluster more than expected (outside two standard deviations) for uniform sampling. The clustering is stronger at small distances and intermediate ones (0–5 cm and around 7–17 cm). We also observed similar clustering for other animals, but most did not exhibit any pronounced spatial arrangement in the kernel density estimate (see Experimental Phase Distributions for more). While not conclusive, these findings provide preliminary evidence that there might be more structure to the arrangements of Hippocampal place cells, and investigation of larger datasets is warranted in the future.

## Discussion

In this work, we have proposed a neural network model that forms place-like spatial representations by decoding learned cognitive maps. Surprisingly, the trained network displays a range of behaviors exhibited by biological place cells, including global remapping across environments and field deformation during geometric manipulations of familiar arenas. Besides reproducing existing place cell experiments, our model makes some surprising predictions.

Our first prediction is that border-type input is sufficient to explain not only place field formation, but also place cell global remapping. While a strong relationship between border cells and place cells has been argued previously (*Barry et al., 2006*; *Hartley et al., 2000*), possible influences on Hippocampal remapping remain relatively unexplored. In our model, we find that place cell remapping arises as a result of sparse input from independent cell assemblies, enacted through strong boundary cell rate changes. Current experimental evidence suggests that border cells largely maintain their firing rate during conditions that elicit place cell remapping (*Solstad et al., 2008*; *Lever et al., 2009*). However, we find that the border code is highly sparse, and so only a small number of such rate-remapping, boundary-type cells would actually be required. Thus, investigating whether border cells *can* display rate changes could be an interesting avenue for future research.

While it could be that border cells in the brain do not (rate) remap, a border-to-place model could still be viable through alternate pathways, such as via gating mechanisms. In this case, a boundary signal projected onto downstream place cells could be gated by contextual signals originating from the lateral Entorhinal Cortex (lEC). *Jeffery, 2011* demonstrated that a gated grid cell input signal could give rise to biologically plausible, place-like spatial signals (*Jeffery, 2011*). In a similar way, gated boundary input could conceivably account for not only place field formation and boundary-selectivity, but also remapping.

Given the range of place cell behaviors our model reproduces, we hold that the border-to-place model it learns should be taken seriously. However, it is worth noting that there are still place behaviors unaccounted for in our work. For instance, we do not observe field doubling when walls are inserted in familiar environments (results not shown), as observed in vivo (*Barry et al., 2006*). However, it is reasonable to suspect that this is due to the lack of sensory information available to the network, as there is no way for the network to detect novel walls. Therefore, adding boundary-selective sensory input to our network could conceivable uncover even more place cell behaviors. This is also supported by the fact that *Uria et al., 2020* observed field doubling in the responses of their model, which utilizes visual input. Thus, adding other sensory modalities may be a fruitful extension of the current model.

A related missing feature is the lack of multiple firing fields as expressed by biological place cells, particularly in large recording environments (*Park et al., 2011*). While our network does exhibit more firing fields when familiar contexts are expanded, place cells can reliably exhibit multiple fields, likely as part of a coding strategy. In contrast, our decoding operation only extracts a single center location, which may limit the expressivity of the network. Future work could, therefore, consider alternative decoding approaches that place even less strict requirements on learned representations.

Our second surprising finding is the model's conspicuous lack of grid cells. As already mentioned, grid cells have been proposed as a possible precursor to place cells (*Jeffery, 2011*; *Moser et al., 2008*; *Solstad et al., 2006*). Grid cells are also often posited as being important for path integration (*Hafting et al., 2005*; *Burak and Fiete, 2009*; *Bush et al., 2015*). Accurate path integration is especially important in the absence of location-specific cues such as landmarks. The only pieces of information available to our model is a velocity signal, an environment-identifying context signal, and a weak, implicit boundary signal (since trajectories cannot exit environment boundaries). As such, there is no explicit sensory information, and path integration is required to solve the position reconstruction task. If grid cells are optimized chiefly for path integration, one would expect that the model learned grid-like solutions. As our model only learns border-type recurrent representations, our findings raise questions concerning the necessity of grid cells for path integration, as well as the causal relationship between place cells and grid cells. That grid cells may not be required to do path integration has also been shown in other recent normative models (*Schøyen et al., 2023*).

While the lack of grid cells in this model is interesting, it does not disqualify grid cells from serving as a neural substrate for path integration. Rather, it suggests that path integration may also be performed by other, non-grid spatial cells, and/or that grid cells may serve additional computational purposes. If grid cells are involved during path integration, our findings indicate that additional tasks and constraints are necessary for learning such representations. This possibility has been explored in recent normative models, in which several constraints have been proposed for learning grid-like solutions. Examples include constraints concerning population vector magnitude, conformal isometry (*Xu et al., 2022*; *Schaeffer et al., 2023*; *Schøyen et al., 2024*), capacity, spatial separation, and path invariance (*Schaeffer et al., 2023*). Another possibility is that grid cells are geared more towards

other cognitive tasks, such as providing a neural metric for space (*Ginosar et al., 2023*; *Pettersen et al., 2024*), or supporting memory and inference-making (*Whittington et al., 2020*). That our model performs path integration without grid cells, and that a myriad of independent constraints are sufficient for grid-like units to emerge in other models, presents strong computational evidence that grid cells are not solely defined by path integration, and that path integration is not only reserved for grid cells.

Besides functional constraints, an important consideration when building neural network models is their architecture. In our model, information primarily flows from recurrently connected, mEC-type units, to CA1-type units by feedforward projections. However, CA1 responses also feed back to the Entorhinal Cortex, via the Subiculum (*Langston et al., 2010*). Such a loop structure is explored in *Morris and Derdikman, 2023*, which also makes use of nongrid spatial cells to inform place field formation, similar to our findings. Incorporating a feedback pathway (from output units to recurrent units) could allow for exploring the connection between grid cells, place cells, and remapping.

While our model does not produce grid-like *representations*, we do observe a striking, grid-like structure in the arrangement of output unit centers. Notably, these centers arrange hexagonally in arenas with open interiors, such as the large square. While a hexagonal placement of field centers has yet to be uncovered experimentally, *Whittington et al., 2020* showed that place cell phases are correlated with grid cell peak locations across environments (*Whittington et al., 2020*). Because the network has learned this particular arrangement to optimize position reconstruction, a hexagonal phase pattern may be optimal for decoding one's position, even in diverse geometries. This is also alluded to by the fact that we observe clustering, and possible structure in the center locations of CA1 place fields in mice data obtained in *Lee et al., 2023*. In the future, larger recordings and in different animals could help solidify whether place field centers exhibit the striking hexagonal arrangement predicted by our model for optimal decoding.

A hexagonal place field arrangement also suggests a possible connection between boundary, place, and grid cells. Boundary-tuned cells could inform place cell pattern formation, which in turn guides grid cell patterns. Such a border-to-place-to-grid cell model could explain grid cell behavior in non-standard or changing environments. For example, grid cells can exhibit (temporary) pattern elongation in novel environments (*Barry et al., 2012*). This grid elongation could be induced by field elongation in place cells, which in turn is caused by boundary field continuation. Besides temporary rescaling, grid patterns are also permanently influenced by environment geometry (*Krupic et al., 2015*), hinting that grid cells receive boundary-dependent input. Furthermore, it has been suggested that border cells serve an error-correcting function for grid cells during navigation (*Hardcastle et al., 2015*). In a boundary-to-place-to-grid model, grid error correction could arise from place-cell inputs informed by boundary responses, or from border cells directly.

In summary, our proposed model, with its notion of a spatial cognitive map and fixed decoding, allows for exploring place cell formation and remapping. In particular, we find that learned place-like representations are formed by boundary input from upstream recurrent units. Global remapping arises from sparse input from differentially activated boundary units. Our work has important implications for understanding Hippocampal remapping, place field formation, as well as the place cell-grid cell system.

## Methods

Code Availability: Code to reproduce models, datasets, and numerical experiments is available from https://github.com/bioAI-Oslo/VPC (copy archived at *Pettersen, 2025*).

## Model & objective

In this work, we trained a recurrent neural network to solve the proposed position reconstruction task *Equation 2* using stochastic gradient descent, using the mean squared error as a loss function, i.e.,

$$\mathcal{L} = \mathbb{E}_t[(\mathbf{x_t} - \hat{\mathbf{x}}_t)^2],$$

where $\mathbf{x}_t$ is a Cartesian target coordinate at a particular time $t$ along a spatial trajectory, and $\hat{\mathbf{x}}_t$ a corresponding predicted position, decoded from the network's output representations (see below).

As the proposed objective function does not impose explicit constraints on the functional form or spatial arrangement of the learned representations, we trained the network in a small set of diverse geometries (see *Figure 1c*) for an illustration. This was done to explore whether the network would learn different representations in different rooms, as a way of optimally encoding the space.

The recurrent network was only given velocity information as input and, therefore, had to learn to perform path integration in order to minimize the position reconstruction task. Concretely, the path integration task consisted of predicting self-position along simulated trajectories. For each trajectory, the network was initialized at a random location in the environment, without initial position information (see 4.2 for initialization details). At every time step $t$ along a trajectory, the network received a Cartesian velocity signal $\mathbf{v}_t$, mimicking biological self-motion information. Denoting a particular point along a trajectory parameterized by time as $\mathbf{x}_t$, the decoded position of the network was

$$\hat{\mathbf{x}}_t = \frac{\sum_{i=1}^N p_i(\mathbf{z}_t)\boldsymbol{\mu}_i}{\max\left(\varepsilon, \sum_{i=1}^N p_i(\mathbf{z}_t)\right)},\tag{5}$$

where $\mathbf{z}_t$ is the network's latent estimate of position at time $t$, formed by integration of previous positions and velocities. In our case, output states $\mathbf{p}_t$ are computed as rectified linear combinations of an upstream recurrent layer (see 4.2 for a description). Meanwhile,

$$\boldsymbol{\mu}_i = \frac{\mathbb{E}_t\left[p_i(\mathbf{z}_t)\mathbf{x}_t\right]}{\max\left(\varepsilon, \mathbb{E}_t\left[p_i(\mathbf{z}_t)\right]\right)} \quad i = 1, 2, 3, ..., N\tag{6}$$

is the center location estimate for output unit $i$, formed using the network states during navigation.

Lastly, we provided the network with a constant one-hot context signal at every timestep, as a token for identifying the environment. The input at time $t$ was, therefore, a concatenation of $\mathbf{v}_t$ and a time-independent context signal $\mathbf{c}$. See *Figure 1d* for an illustration.

## Neural network architecture and training

In this work, we consider a one-layer vanilla recurrent neural network (RNN) featuring $N_g = 500$ units. These recurrent units project linearly onto an output layer consisting of $N_p = 100$ units. Both recurrent and output units were equipped with ReLU activation functions and no added bias.

At time $t$, the hidden state of the recurrent layer was given by

$$\mathbf{g}_{t+1} = \left[W_R\mathbf{g}_t + W_I\mathbf{I}_t\right]_+,\tag{7}$$

where $W_R \in \mathbb{R}^{N_g \times N_g}$ is a trainable matrix of recurrent weights, and $W_I \in \mathbb{R}^{N_g \times N_I}$ a matrix of input weights, with $\mathbf{I}_t$ being the input at time $t$, and $N_I$ the dimensionality of the input signal. The input consisted of a concatenation of a velocity signal and a six-entry, one-hot context signal, i.e., $\mathbf{I}_t = \mathrm{cat}(\dot{\mathbf{x}}_t, \mathbf{c})$. Subsequently, output states were computed according to

$$\mathbf{p}_t = [W_p\mathbf{g}_t]_+,$$

where $W_p \in \mathbb{R}^{N_p \times N_g}$ is a trainable weight matrix.

Feedforward weights were all initialized according to a uniform distribution $\mathcal{U}(-k_i, k_i)$, where $k_i = 1/\sqrt{N_i}$ with $N_i$ being the number of inputs to that layer. For the recurrent layer, the RNN weight matrix was initialized to the identity. This was done to mitigate vanishing/exploding gradients caused by the long sequence lengths used for training, as suggested by *Le et al., 2015*.

To explore network dynamics when transitioning between different environments, we trained the recurrent network in a *stateful* fashion. This involved maintaining the recurrent state from the end of one trajectory, and using it as the initial state along a new trajectory. For each transition, the new environment and the starting location within that environment were sampled randomly (and uniformly). To ensure stability, the network state was reset every ten trajectories, to an all-zero initial state. When reset, the network state was initially set to all zeros, providing no positional information at the start of the trajectory. While the network state was carried between different environments, gradient calculations were truncated at the end of each episode.

Because the network is not provided with initial position information (all-zero initial state), the network has to infer its location within an environment (the identity of which is known due to the

context-input) based on its geometry, e.g., through border interactions. This requires a large sample (long trajectory) of the geometry. The recurrent network was, therefore, trained on trajectories of sequence length $T = 500$. The minibatch size used for gradient descent was 64. Because of statefulness and resetting, the network, therefore, experienced effective sequences of 5000 timesteps during training. However, no gradient information was carried between subsequent trajectories.

To implement models, we used the PyTorch Python library (*Paszke, 2019*). We used the Adam optimizer (*Kingma, 2017*) for training, with a learning rate of $10^{-4}$ and otherwise default parameters (*Paszke, 2019*). The network was trained for a total of 100 epochs using the training dataset detailed in 4.3. To regularize the network, we applied an L1 penalty to the recurrent network states, i.e., **g**. The associated L1 hyperparameter $\lambda$ was set to 10.

## Trajectory simulation and datasets

Networks were trained using simulated datasets of trajectories traversing 2D geometries. The starting location of a trajectory was sampled randomly and uniformly within an environment. To sample points uniformly from non-square geometries, a rejection sampling strategy was used: First, points were sampled according to a uniform distribution, whose support was given by the smallest rectangle that completely covered the given geometry. Then, ray casting was done to determine whether points were in the interior of the geometry. Concretely, a horizontal ray was cast from a given point and the number of intersections with the enclosure walls was determined. If the number of intersections was odd, the point was accepted as being inside the environment. If the number of intersections was even, the point was resampled. This procedure was iterated until the desired number of samples was obtained. Note that the interior determination method only works for extended objects, such as holes. Therefore, to add thin environment boundaries (infinitely thin walls), we simply superimposed two boundaries with no spatial separation.

To approximate the semi-smooth motions of foraging rodents, trajectory steps were generated by drawing step sizes according to a Rayleigh distribution with $\sigma = 0.5$, and heading direction from a von Mises distribution centered at the previous heading with scale parameter $\kappa = 4$. To ensure that the random walk remained within the geometry, we checked whether a proposed step intersected with any of the environment walls. If an intersection was detected, the heading direction was resampled until an allowed step was achieved. This procedure was iterated until the desired amount of timesteps was obtained. Note that step sizes were not resampled. This procedure yields smooth trajectories, with inherent turning away from boundaries. Trajectory positions were generated using a forward Euler integration scheme, with timestep $dt = 0.1$.

For computational efficiency, the network was trained and evaluated on precomputed datasets of trajectories. The full dataset contained 15000 trajectories, each of which was 500 timesteps long. Of these, 80% was reserved for training, and the remaining 20% for validation. In both datasets, an equal number of samples were included for every environment. All analyses were conducted using newly generated test trajectories.

## Contexts and geometries

To explore the possibility that the model could learn to remap, we trained networks in multiple distinct environments, each labeled by a unique, one-hot context vector. The included geometries were square, circular, and rectangular. In addition, we also included a large square, a square with a thin, central dividing wall, and finally, a square with a central hole. Each geometry and associated context signal is illustrated in *Figure 1c*.

## Remapping experiments

We conducted two remapping experiments to study whether the behavior of the trained neural networks aligned with those observed experimentally in animals. The first consisted of running the trained network (with frozen weights) in multiple familiar geometries (with the corresponding context signal), similar to canonical remapping experiments (*Leutgeb et al., 2004*; *Fyhn et al., 2007*). Referring to *Figure 3a* we ran the trained recurrent network along 25,000-timestep sequences, that initially visited the square environment. Then, the network was transferred to the square with a central wall, before being returned to the square environment. For each trajectory, the starting

position was sampled randomly within the geometry, and the state of the network was maintained between environments. The initial state of the network in the first environment was set to the zero vector.

The second set of experiments was designed to explore the consequences of geometric manipulations of familiar environments on the learned spatial representations. To do so, we ran the trained network (with fixed weights) in elongated versions of the familiar environments, similar to the experimental setup in *O'Keefe and Burgess, 1996*.

The first of these trials involved running the trained RNN in the square environment, with the appropriate context. However, during inference the environment walls were elongated by factors of 2 and 3 compared to their original length. For reference, *Figure 4a* illustrates the environment rescaling protocol.

The second trial concerned the effects of extending a familiar environment to previously unseen locations. Concretely, this experiment entailed transforming the environment with a central hole, into a square environment, while retaining the original context signal. In other words, the four walls of the central hole were removed, allowing movement in previously inaccessible parts of the arena. The third trial featured rescaling of the square environment into a larger square, i.e., proportional scaling in both horizontal and vertical directions while maintaining the context cue of the square environment. Again, wall lengths were scaled by factors of 2 and 3.

The final geometric manipulation involved expanding the circular environment uniformly, again by factors of 2 and 3, respectively.

## Attractor dynamics and noise injection

To investigate whether the learned representations exhibited attractor-like behavior, we performed a noise-injection experiment. The experiment consisted of evaluating the trained RNN on 1000, 800-timestep trajectories within the square environment. At the midpoint of the trajectory Gaussian noise was injected into the recurrent state. This perturbed state was subsequently rectified, before the network was run normally for the remainder of the trajectory. We performed the same experiment for multiple noise levels $\sigma = \{0.0, 0.01, 0.1, 1\}$, where $\sigma$ determines the scale of the normal distribution used for noise generation. The state of the RNN directly after noise injection could, therefore, be described as $\mathbf{g}_{\text{perturbed},i} = [\mathbf{g}_\tau + \chi_i]_+$, where $\chi$ is a vector of random variables, drawn from a multivariate normal distribution. $\tau$ denotes the time of noise injection, taken to be timestep 400, while $[\cdot]_+$ is a rectification operation.

To assess whether the representation was stable, and whether the state of the network was attractive, we computed ratemap population vector correlations (see 5 for details) between every time point in the sequence for each noise level.

## Low-dimensional representations and explainability

To better understand the behavior of the recurrent network, we performed PCA, alongside dimensionality reduction using UMAP (*McInnes et al., 2020*). PCA was done on the recurrent and output states of the network as it was run on long (10000 timesteps in each environment) trajectories that visited every environment sequentially. For each environment transition, the state of the network was maintained. PCA was performed for each environment separately, as well as for the full trajectory visiting every environment. As an example, for a trajectory of length $T$, the output activity $\mathbf{p} \in \mathbb{R}^{T,N_p}$ was projected to a low-dimensional representation $\tilde{\mathbf{p}} \in \mathbb{R}^{T \times n_{pca}}$, with $n_{pca}$ being the number of principal components.

The second dimensionality reduction technique consisted of performing UMAP (*McInnes et al., 2020*) on the states of the network along the full trajectory. This was done to explore whether network activity resided on a low-dimensional manifold. Population activity at each time point was subsequently projected down to three dimensions, yielding a dimensionality-reduced vector representing the full network activity at a particular point along the trajectory. To further explore the dynamics of the network, we computed the eigenvalue spectrum of the recurrent weight matrix. Finally, we computed Pearson correlation coefficients between columns of the input weight matrix corresponding to different context signals.

## Analyses

To compare the representational similarity of the network output across environments and time, we performed several analyses using unit ratemaps.

## Ratemaps

Ratemaps of unit activity were computed by discretizing environments into bins. The rate was then determined by dividing unit activity by the number of visitations to that bin along a single trajectory. Unless otherwise specified, ratemaps were formed using 25,000-timestep trajectories. For long-sequence experiments, a burn-in period of 500 initial timesteps was excluded from ratemap creation. This was done to only include the steady-state behavior of the network. For the remapping dynamics in *Figure 3e*, ratemaps were created by aggregating responses over 500 distinct, 800-timestep trajectories.

## Spatial correlation

Following (*Fyhn et al., 2007*), we computed unit-wise ratemap spatial correlations to investigate possible remapping behavior. For a single unit, the spatial correlation was calculated by computing the Pearson correlation coefficient between flattened unit ratemaps. We considered the correlations between the square environment, and the square with a central wall, due to their geometric similarity. In other words, the ratemap of a unit in the square environment was correlated with its ratemap in the square with a central wall environment. This procedure was repeated for all units that were active (exhibited nonzero activity) in both environments, and a distribution of spatial correlations was formed. As a baseline, a shuffled distribution was computed by correlating every active unit with every other active unit, across environments. Finally, correlations were computed for relative ratemap rotations of 0, 90, 180, and 270 degrees, and the maximal correlation reported. This was done to account for the possibility that remapping consisted of a rigid rotation in space.

## Ratemap population vector correlation

To compare the representational similarity of entire unit populations at different timepoints (as in *Figure 5*), we computed the Pearson correlation between ratemap population vectors at different times. A ratemap population vector at a particular time was constructed by stacking the flattened ratemaps of every unit into a single array of dimension $N_{units} \cdot N_x \cdot N_y$ with $N_{units}$ being the number of units in the relevant layer, and $N_x = N_y = 16$ is the number of bins along the canonical $x, y$ directions. In other words, this measures the correlation of an entire population at every spatial point, at a particular point in time, and can thus be used to gauge the time evolution of an entire representation. Note that states were aggregated across multiple trajectories, at the same time point.

Using Astropy (*Price Whelan et al., 2022*), Gaussian smoothing with NaN interpolation was used to fill in unvisited regions. The smoothing kernel standard deviation was one pixel. For the experiment featuring transfers between different environments (*Figure 3e*), only units with nonzero activity in one or more environments were included in the population vector.

## Rate overlap and difference

As a measure of rate changes between conditions, we computed the rate overlap (*Leutgeb et al., 2004*), and rate difference. Considering two conditions (e.g. comparing across two environments), rate overlap was computed by dividing the mean activity in the least active condition by that in the most active. Only units that were active in at least one condition were included in the analysis. The rate difference was computed by simply subtracting the activity in one condition from that in another, and dividing by the sum of activity in both conditions. This measure is similar to the rate difference used in *Leutgeb et al., 2005*, but maintains the sign of the difference. As with the rate overlap, only units that were active in at least one condition were included.

For both the overlap and difference, a shuffled distribution was formed by randomly pairing units across conditions. For both quantities, pairings were performed 1000 times.

## Spatial information

To select the most place-like units for the phase distribution visualization, we computed the spatial information content (*Skaggs and McNaughton, 1992*) of all units. Using unit ratemaps of $M$ bins, the spatial information of a single unit was computed as

$$S = \sum_i^M f_i \bar{f} \log_2 \frac{f_i}{\bar{f}} p_i,$$

where $p_i$ is the occupancy of bin $i$, $f_i$ is the firing rate in that bin, while $\bar{f}$ is the unit's average firing rate over all bins. High spatial information units were subsequently selected as those whose spatial information were above the 2.5th percentile in all environments.

## Ripley's H and clustering

To assess whether biological place fields exhibit non-uniform clustering, we computed the Ripley's H statistic (*Lagache et al., 2013*) for the center locations of place cells recorded from animals (*Lee et al., 2023*).

For a set of $N$ points, we computed Ripley's H in two steps: First, we determined Ripley's K, which counts the average number of points within a distance $R$ of a point, given by

$$K(N, R) = \frac{|\Omega|}{N(N-1)} \sum_{\mathbf{x} \neq \mathbf{y}} \mathbf{1}_{\{|\mathbf{x}-\mathbf{y}|<R\}} f(\mathbf{x}, \mathbf{y}),$$

where $\mathbf{x}$ and $\mathbf{y}$ are distinct points, $|\Omega|$ is the area of the domain $\Omega$ encompassing the set of points, while $\mathbf{1}$ is the indicator function. $f(\mathbf{x}, \mathbf{y})$ is a boundary correction factor to account for a lack of observations outside the region $\Omega$. We follow *Lagache et al., 2013* and take

$$f(\mathbf{x}, \mathbf{y}) = \frac{1}{2}(k(\mathbf{x}, \mathbf{y}) + k(\mathbf{y}, \mathbf{x})), \text{ with}$$

$$k(\mathbf{x}, \mathbf{y}) = \frac{|\partial b(\mathbf{x}, |\mathbf{x} - \mathbf{y}|)|}{|\partial b(\mathbf{x}, |\mathbf{x} - \mathbf{y}|) \cap \Omega|},$$

where $\partial b(\mathbf{x}, |\mathbf{x} - \mathbf{y}|)$ is the circumference of a ball centered at $\mathbf{x}$ of radius $|\mathbf{x} - \mathbf{y}|$, and the denominator the circumference of the part of the ball that is inside the geometry. We used the Shapely Python library (*Gillies et al., 2024*) for computing intersections between balls and the enclosing geometry. The second step consisted of centering and normalizing Ripley's K, obtaining Ripley's H, given by

$$H(N, R) = \sqrt{\frac{K(N, R)}{\pi}} - r.$$

For our analysis, we computed Ripley's H for center locations of place cells in mice traversing a 75×75 cm square environment (*Lee et al., 2023*) over four distinct recording days. Centers, in this case, were decoded as the maximum firing location in 15×15 smoothed ratemaps. For each animal, we included place cells above the 75th percentile. For each cell, the ratemap corresponding to the recording day with the largest spatial information was selected. As a baseline, Ripley's H was computed for 100 sets of points sampled randomly and uniformly on a 15×15 square grid, matching the spatial discretization of the ratemaps used. For both baseline and real data, ball radii were varied from $\varepsilon = 10^{-8}$ to approximately 26.5 cm, corresponding to a quarter of the square's diagonal.

To visualize possible clustering of place fields, we computed Gaussian kernel density estimates of decoded field centers. This procedure was repeated for all animals, and only centers of cells with spatial information above the 75th percentile were included. For all kernel density estimates, the bandwidth parameter was set to 0.2, and kernels were evaluated on 64×64 grids. See (*Lee et al., 2023*) for details on ratemap creation and experiments.

## Acknowledgements

We would like to thank J Quinn Lee and Mark Brandon of McGill University, as well as their co-authors, for graciously sharing their data with us. We hope others follow their example of open and helpful collaboration.

## Additional information

### Funding

| Funder | Grant reference number | Author |
|---|---|---|
| Research Council of Norway | | Anders Malthe-Sørenssen Mikkel E Lepperød |

The funders had no role in study design, data collection and interpretation, or the decision to submit the work for publication.

### Author contributions

Markus Borud Pettersen, Conceptualization, Data curation, Software, Formal analysis, Validation, Investigation, Visualization, Methodology, Writing – original draft, Writing – review and editing; Vemund Schøyen, Conceptualization, Data curation, Software, Supervision, Investigation, Visualization, Methodology, Writing – original draft, Project administration, Writing – review and editing; Anders Malthe-Sørenssen, Conceptualization, Resources, Supervision, Funding acquisition, Writing – original draft, Project administration, Writing – review and editing; Mikkel E Lepperød, Conceptualization, Resources, Data curation, Formal analysis, Supervision, Funding acquisition, Validation, Investigation, Visualization, Methodology, Writing – original draft, Project administration, Writing – review and editing

### Author ORCIDs

Markus Borud Pettersen ⓘ http://orcid.org/0000-0001-9004-4995
Vemund Schøyen ⓘ http://orcid.org/0000-0001-8932-5706
Anders Malthe-Sørenssen ⓘ http://orcid.org/0000-0001-8138-3995
Mikkel E Lepperød ⓘ https://orcid.org/0000-0002-4262-5549

Reviewer #1 (Public review): https://doi.org/10.7554/eLife.99302.4.sa1
Reviewer #2 (Public review): https://doi.org/10.7554/eLife.99302.4.sa2
Reviewer #3 (Public review): https://doi.org/10.7554/eLife.99302.4.sa3
Author response https://doi.org/10.7554/eLife.99302.4.sa4

### Data availability

The current manuscript is a computational study, so no data has been generated for this manuscript. Modelling code is found at https://github.com/bioai-oslo/vpc (copy archived at *Pettersen, 2025*).

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

# Appendix 1

## Long sequence evaluation

To verify that the model performs accurate path integration, even for very long sequences, we computed Gaussian kernel density estimates of the Euclidean decoding error at every step along 100, 10000 timestep trajectories in each environment. The bandwidth parameter was set to approximately 0.4 according to Scott's Rule, and the resulting error distributions are shown in *Appendix 1—figure 1*.

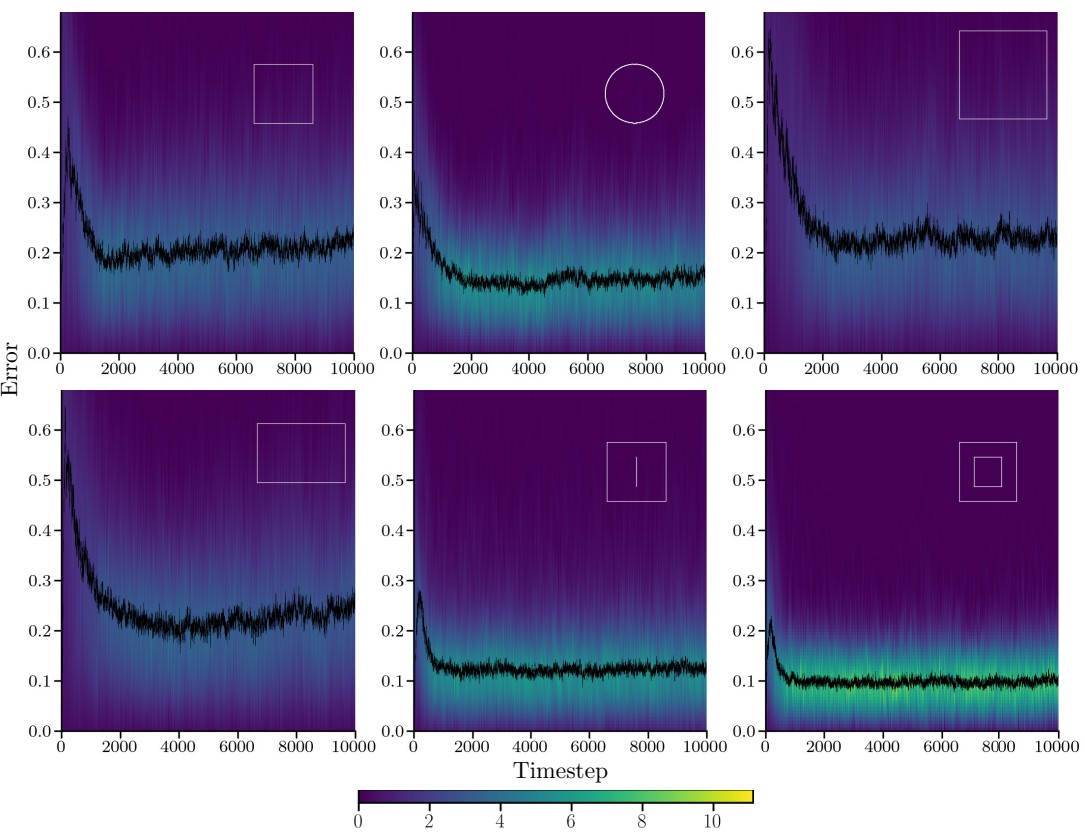

**Appendix 1—figure 1.** Error distribution for long sequence evaluation. Each pane shows the distribution and median of Euclidean distances (error) between true and decoded trajectories for the trained recurrent neural network (RNN) evaluated on 100 long (10000-timestep) test trajectories in a particular environment (inset). The color indicates the kernel density estimate value at a particular timestep.

## Extended model ratemaps

*Appendix 1—figures 2 and 3* show ratemaps for all 100 output units and 100 recurrent units, respectively. Responses in every environment are included. Notably, both output and recurrent units are sparse, with most recurrent units silent in a given environment. Output units display field shifts between environments, indicative of remapping.

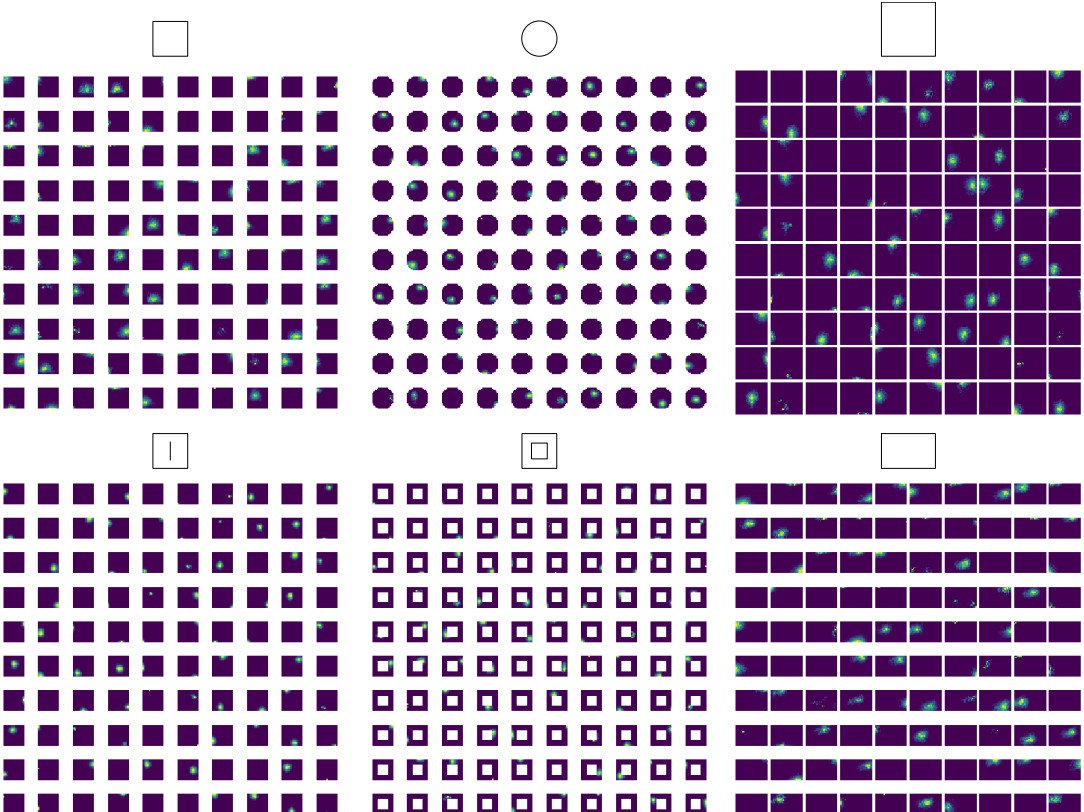

**Appendix 1—figure 2.** Ratemaps of all 100 output units in each environment. The geometry is indicated atop every ensemble. Unit identity is given by its location on the grid (e.g. unit 1 is top left in each environment).

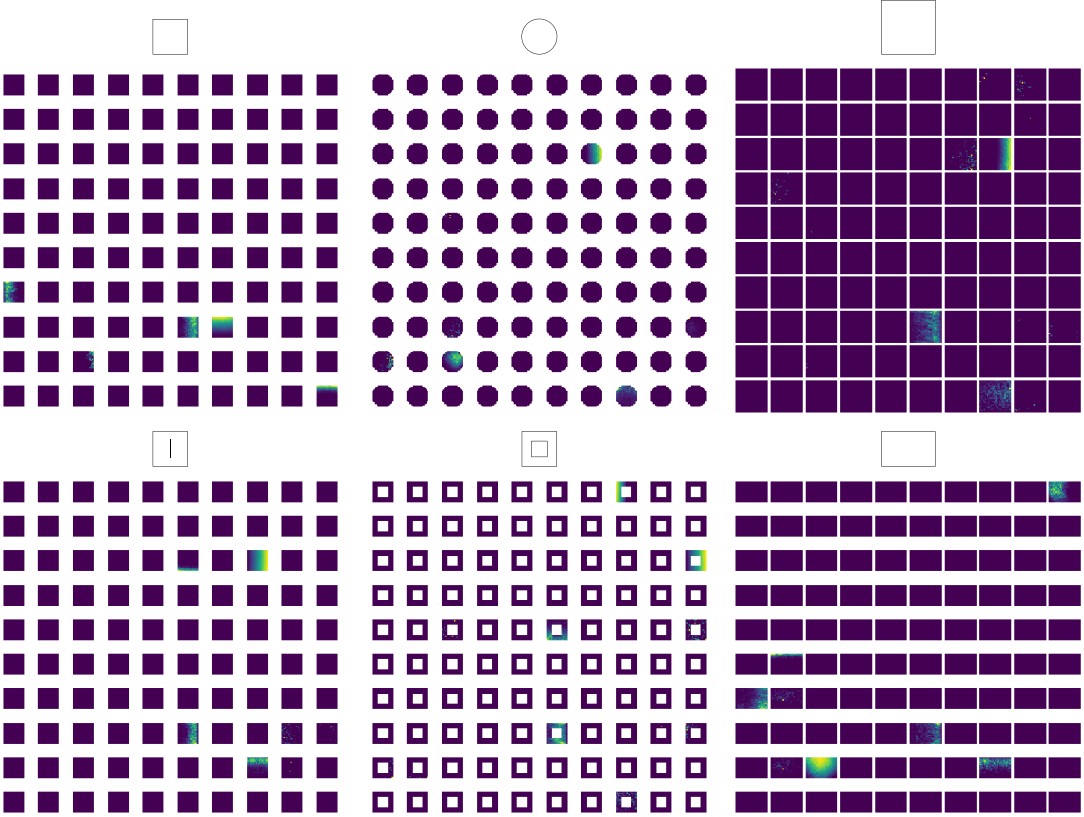

**Appendix 1—figure 3.** Ratemaps of 100 recurrent units in each environment. The geometry is indicated atop every ensemble. Unit identity is given by its location on the grid (e.g. unit 1 is top left in each environment).

## Experimental phase distributions

Figure *Appendix 1—figure 4* shows example ratemaps, animal trajectories, and estimated distributions of center locations, for centers decoded from ratemaps of high spatial-information place cells in mice (data provided by *Lee et al., 2023*). Also shown is Ripley's H for center locations decoded from the same cell ratemaps.

While some distributions display no clear patterns in their center arrangements (e.g., for animal QLAK-CA1-74), some distributions do display signs of clustering and center locations even show some signs of regularity in their arrangement (e.g., QLAK-CA1-50), possibly even similar to the hexagonal arrangements we observe in our model (as shown in *Figure 7*).

To quantify whether the ensemble of place cells were indeed hexagonally arranged, we computed the grid score *Sargolini et al., 2006* of a kernel density estimate over decoding center locations. In this case, all center distributions exhibited low grid scores. However, as it results from correlating rotated versions of an autocorrelogram, the grid score is sensitive to differential clustering in the place cell center distribution. Such clustering could occur in data due to e.g. insufficient sampling of cells encoding particular spatial locations. To combat this possibility, we extracted the peaks of the center location KDE, and computed the grid score of the corresponding KDE over these peaks. The resulting peak grid score measures the degree of hexagonality in the distribution while smoothing out differences due to unequal clustering.

This analysis revealed that many animals displayed no clear orientation in their center distribution (e.g. QLAK-CA1-08). However, some exhibited exhibited a tendency towards some degree of hexagonal symmetry (QLAK-CA1-50; peak GS 0.67, QLAK-CA1-75; peak GS 0.27). Notably, these are also the two animals with the largest number of included cells.

Another interesting finding, is that most animals exhibited stronger clustering than expected by random (and uniform) center distributions (as evidenced by greater-than-random Ripley's H for small

ball radii). Note however, that this result could reflect the clustering of field centers near boundaries, which is seen in almost all included animals.

While preliminary, these findings, in light of our computational predictions, warrant further investigation into the distributional properties of place cells center locations in even larger, high quality datasets.

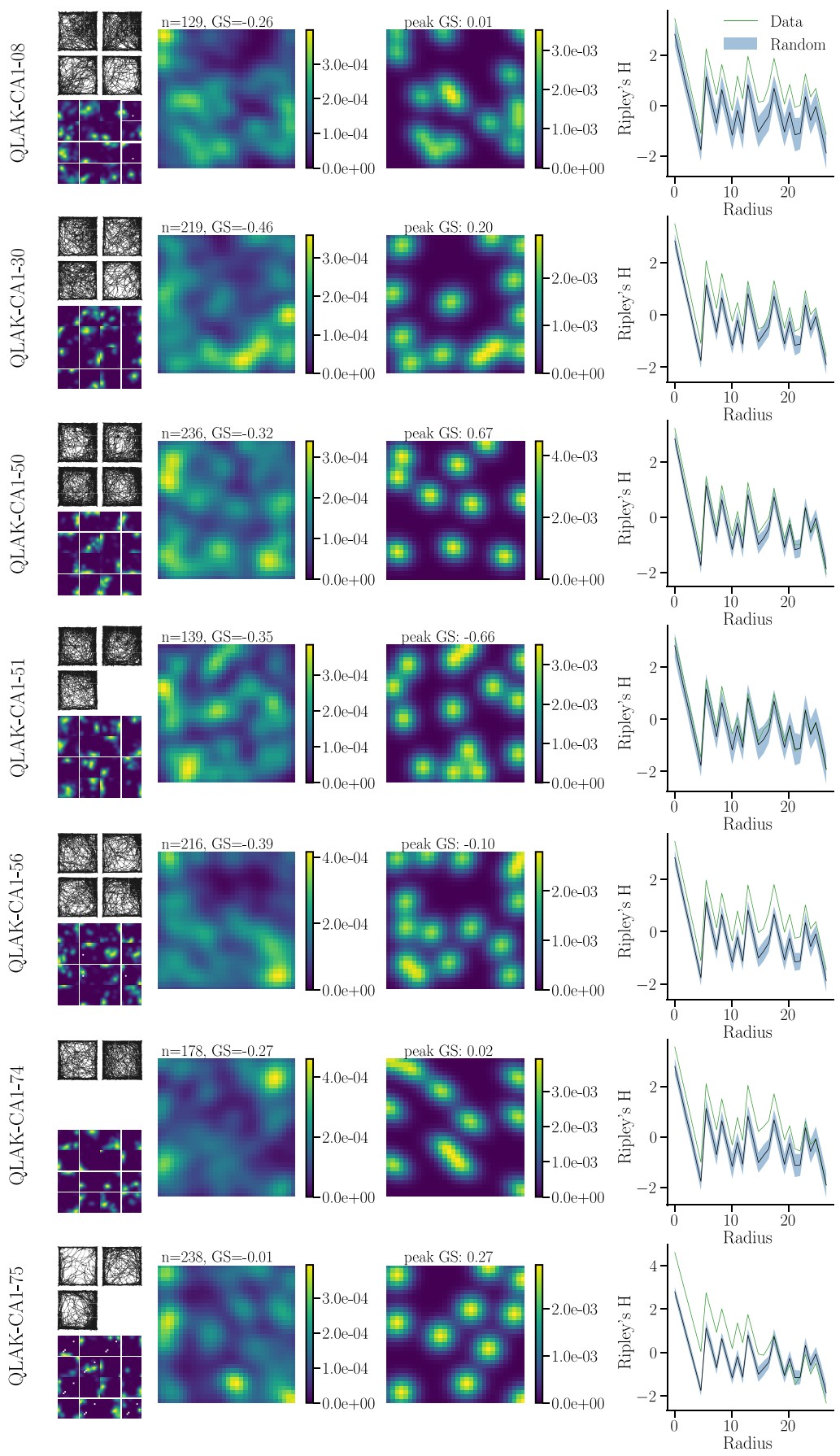

**Appendix 1—figure 4.** Place cell center distributions in mice. From left to right: trajectories, as well as example ratemaps of CA1 place cells in mice (animal indicated by title), next to kernel density estimates of field center locations (locations of maximal firing rate) for cells with high spatial information. Inset is the number of included cells ($n$) and grid score of the KDE. Also shown is a KDE over maximum locations, alongside the corresponding maximum location grid score (peak GS). The rightmost panel shows Ripley's H for all place cell peak locations.

## Velocity ablation

In 5, we investigated whether the network exhibited attractor-like behavior in the face of noise injection. However, we only considered the case where the network was allowed to explore the arena (with velocity input) after noise injection. In this case, it is possible that representations converge back to their original states after performing error-corrections based on boundary interactions in the environment. This is not entirely consistent with the notion of an attractor state. Therefore, we repeated the noise-injection experiment, with velocities ablated upon noise injection. In other words, the network was allowed to run for an extended period of time ($t = 399$ steps) before noise was injected into the network state.

The resulting population vector correlations are shown in *Appendix 1—figure 5a, b*. As seemingly expected, state-to-state correlations are approximately constant over time in the zero velocity, zero-noise case ($\sigma = 0.0$). However, for increased noise, output unit correlations become smaller ($\sigma = 0.01$), and for large noise levels, output states before and after noise injection become decorrelated. As seen in *Appendix 1—figure 5c*, larger noise levels coincide with output unit silencing ($\sigma = 0.1$), while some units appear to shift firing location ($\sigma = 1.0$). Together, these findings suggest that for small noise levels, network representations do converge to their steady-state value ($\sigma = 0.01$). However, for sufficiently large noise, network states fail to converge. In 5 on the other hand, representations converge even for the largest noise scales. Thus, while the network exhibits attractor-like behavior for low noise levels, it seems that exploration can serve as an error-correcting mechanism for larger noise scales.

An interesting aspect of this result, is that recurrent representations remain somewhat correlated, even for maximal noise injection (*Appendix 1—figure 5b*), even when output units are uncorrelated. This could suggest that recurrent units, while maintaining some of their spatial tuning, fire insufficiently to drive output units beyond their firing threshold (which is zero, due to the ReLU activation function). Another important point is the size of the injected noise. *Appendix 1—figure 5d* showcases that noise levels are large relative to typical recurrent unit firing rate, especially considering that most recurrent units are silent. Considering that the network has not experienced any noise during training, it is surprising that representations are quite robust to noise, with or without a velocity signal. Collectively, these findings all support the notion that the network has learned attractor-like structures corresponding to the explored environments.

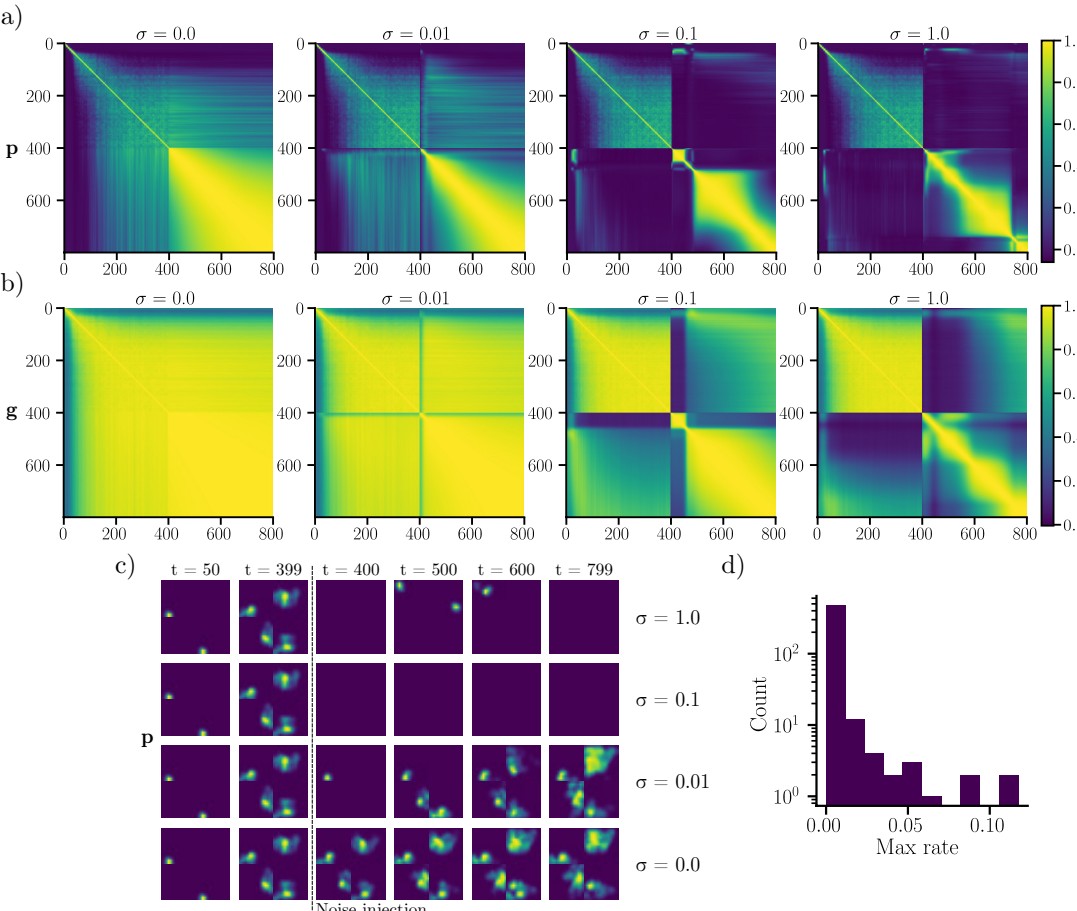

**Appendix 1—figure 5.** Representational stability after velocity ablation. (**a**) Output unit ratemap population vector correlations between all timesteps for trajectories visiting the square environment, with noise injected at timestep.$t = 400$ Velocity input is ablated after noise injection. (**b**) As in (**a**), but for recurrent units. (**c**) Ratemaps of output units before and after noise injection, for varying noise scales.$\sigma$ (**d**) Distribution of maximum firing rates for all recurrent units across all timesteps, when no noise is injected.

## Learned recurrent representations align with environment boundaries

One peculiar feature of the learned, border-like representations exhibited by the recurrent layer, is their propensity to align with the cardinal directions. In most arenas, this coincides with the orientation of environment boundaries. However, to some extent, cardinal spatial tuning is also evident in the only non-polarized environment, i.e., the circular arena (see e.g. *Figure 4d*), where recurrent responses are slightly rotated, but mostly align with the horizontal and vertical directions. This raises the question of whether recurrent units could inherit their tuning profile from some source other than the environment geometry, such as the Euclidean coordinate system of the velocity input signal.

To explore whether recurrent unit responses actually align with environment boundaries, we trained a new RNN model in a rotated square (rhombus) environment, while keeping all other training parameters fixed (see Neural Network Architecture and Training for details). Ratemaps of recurrent unit responses for this network is shown in *Appendix 1—figure 6*. In this case, recurrent responses are still highly sparse, as most units are silent. However, as with the multi-environment model, some units display boundary-like tuning. As expected from boundary-tuned units, responses are no longer aligned with the cardinal directions, but rather with the walls of the arena, providing evidence that representations are indeed tuned to boundaries. This also aligns with the fact that no positional information is available to the recurrent network, and that self-localization must rely on boundary interactions.

However, the apparent polarization of representations in the circular environment could still point to several interesting possibilities: For one, it might be that responses in the circular environment inherit some of their tuning from learning in other, polarized environments, which might also facilitate representational reuse across environments. As an alternative, representational polarization may also reflect the coordinate system of the velocity signal, in the absence of environmental directionality. Exploring more biologically inspired inputs, such as introducing head direction units, could therefore prove an interesting extension of the current model.

In summary, learned recurrent representations appear to be tuned to the boundaries of the environment, but investigating how and why such a firing profile emerges could prove an interesting avenue for future work.

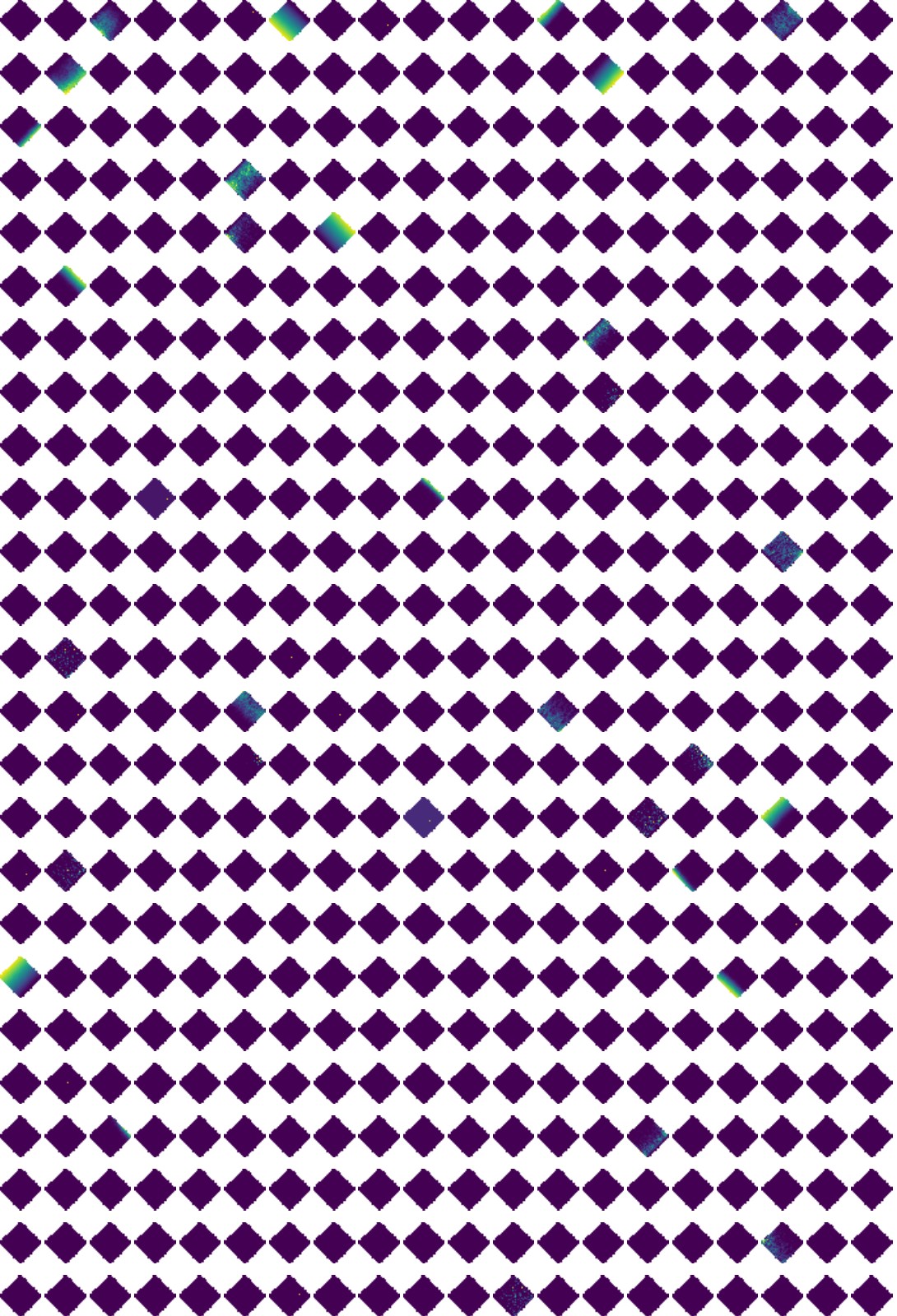

**Appendix 1—figure 6.** Recurrent representations in a rhombus environment. Spatial representations of all 500 recurrent units for a model trained in a rotated square (rhombus) environment.

**Representations are guided by context**

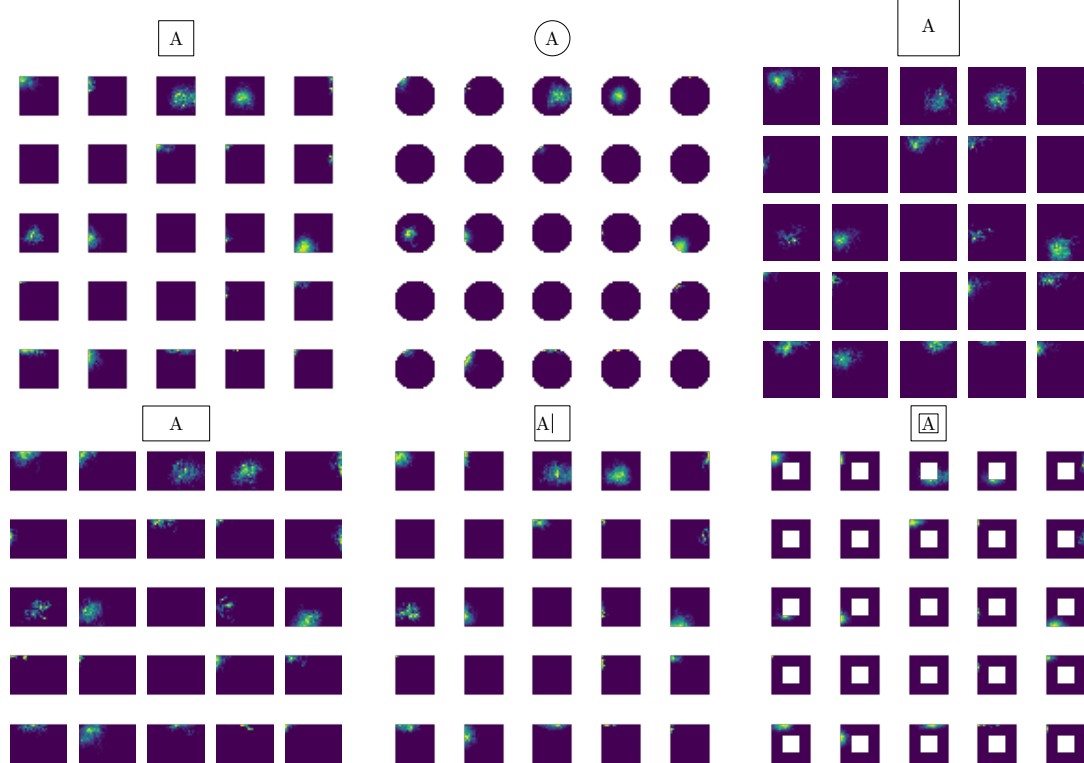

**Appendix 1—figure 7.** Ratemaps during context-geometry mismatch. Ratemaps of 25 randomly selected output units, evaluated in all geometries, when the context signal is fixed equal to the square context signal (**A**).

We have established that the network exhibits attractor-like behavior in the face of injected noise. The question then remains how the network selects a particular attractor state to converge towards. In particular, there are two possibilities that appear particularly likely; the context signal, or the geometry of the environment. To distinguish these two cases, we conducted a simple context mismatch experiment, by evaluating the trained network in all geometries, while keeping the context signal fixed at the value corresponding to the square arena. Resulting ratemaps are showcased in *Appendix 1—figure 7*, which demonstrates that representations in all geometries coincide with those in the original square environment. This indicates that the network relies on the context signal alone for attractor state selection. Note that in large environments, like the rectangle or large square, firing fields appear somewhat enlarged (consistent with 4). Thus, representations are responsive to changes in the geometry, but the global remapping-type behavior between contexts appears to be determined by the context signal itself.

## Recurrent units rate remap

We have demonstrated that output units undergo global-type remapping between distinct contexts. However, this remapping behavior appears to be facilitated by rate remapping in recurrent units. This is shown explicitly in *Appendix 1—figure 8*, which includes rate differences between recurrent responses across all environments. Notably, responses appear highly independent across environments. This can be seen from the large proportion of (normalized) rate differences of either −1 (unit becomes silent), or +1 (unit becomes active). Thus, recurrent unit exhibits strong rate remapping, and the recurrent code is highly sparse when comparing across environments.

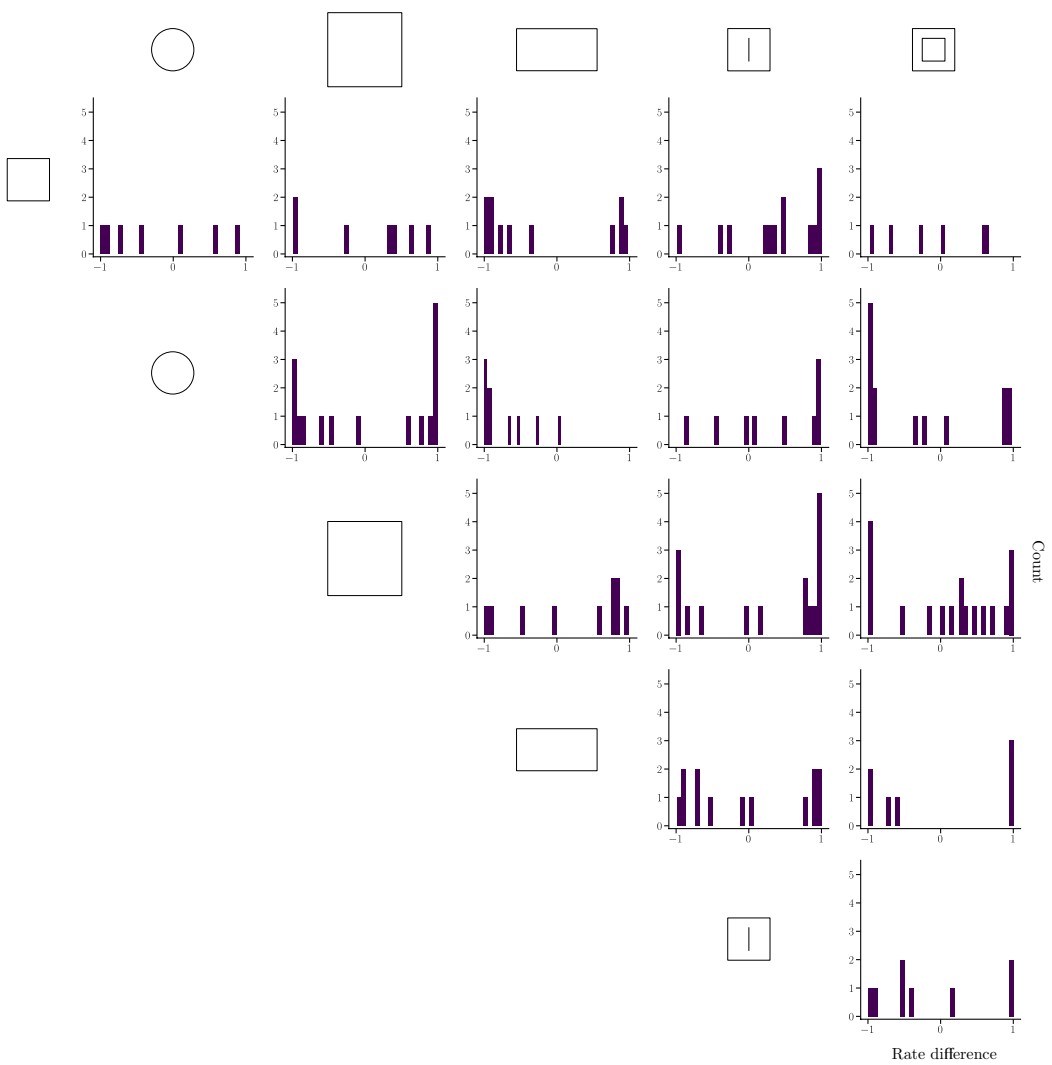

**Appendix 1—figure 8.** Rate differences for recurrent units across all environments. Included are differences when comparing rates between the environment illustrated on the diagonal and all other environents (columns; illustrated in top inset). Note that differences are (anti-)symmetric.

## A taxonomy of cognitive maps

With the definition of a cognitive map in *Equation 1*, we can categorise and compare recent normative neural navigation models. A range of models have recently been put forward that solve tasks similar to ours. In this section, we provide a brief recap of these models, and show that they may be viewed as instances of the cognitive map in *Equation 1* with different constraints and target representations.

Common to these models is that they all make use of random sampling of space in the form of simulated spatial trajectories in bounded 2D spaces, motivated by the foraging behaviour of rats. In addition, most works employ gradient-based optimization schemes, and optimize over independent minibatches. We will, however, omit indexing by minibatches for brevity.

For example, *Dordek et al., 2016* used a target representation $\mathbf{u}(\mathbf{r}) = \mathbf{p}(\mathbf{r})$ of place cells modelled as either zero-mean Gaussians or difference of Gaussians, with $\mathbf{r}$ being a Cartesian coordinate which is encoded into a target place code. The target unit centre was sampled from a random, uniform distribution.

The task, in this case, was to perform non-negative PCA on the label place cells in a square domain $\Omega$, i.e., finding a constrained low-dimensional representation of the label activity. Concretely, we can formulate PCA as the minimization problem

$$\mathcal{L} = \|\mathbf{p}(\mathbf{r}) - \hat{\mathbf{p}}(\mathbf{r})\|^2$$
$$\text{s.t.} \quad W^T W = I_N, \quad W_{ij} \geq 0, \tag{A8}$$

with $W \in \mathbb{R}^{M \times N}$, $M \leq N$ and where $\hat{\mathbf{u}}(\mathbf{r}) = \hat{\mathbf{g}} \cup \hat{\mathbf{p}}$, $\hat{\mathbf{g}} = W\mathbf{p}$ and $\hat{\mathbf{p}} = W^T \hat{\mathbf{g}}$. The authors found that grid-like responses $\hat{\mathbf{g}}$ appear as an optimal low-dimensional representation of the target place code $\mathbf{p}$. This formulation *Dordek et al., 2016* is suitable for studying optimal cognitive maps in an idealized spatial setting. In the ideal setting, the candidate map is learned directly from true spatial coordinates, in contrast to the case where this information is latent, and agents have to build estimates of their location by integrating several sources of spatial information, such as landmark locations and path integration.

*Cueva and Wei, 2018*, *Banino et al., 2018*, *Sorscher et al., 2023* learns latent spatial representations through path integration in a recurrent neural network model. The state of the recurrent network at time $t$ is given by a recurrence relation

$$\hat{\mathbf{u}}_{t+\Delta t} = \hat{\mathbf{u}}_{t+\Delta t}(\hat{\mathbf{u}}_t, \mathbf{v}(t), \Theta), \tag{A9}$$

where $\Theta$ denotes a set of model parameters, and $\mathbf{v}(t)$ the input velocities at some time $t$, while $\Delta t$ is an increment of time. For the RNNs described in the coming sections, we suppress the dependency on parameters $\Theta$ for the sake of readability.

*Cueva and Wei, 2018* considered a version of the cognitive map *Equation 1* in which a recurrent neural network was trained to minimize the reconstruction error and soft constraints

$$\mathcal{L} = \|\mathbf{r}_t - \hat{\mathbf{r}}_t\|_2^2,$$
$$\mathcal{C}_1 = \lambda_1 \sum_t |\hat{\mathbf{g}}_t|_2^2, \quad \mathcal{C}_2 = \lambda_2 \|W_{in}\|_F{}^2 \quad \mathcal{C}_3 = \lambda_3 \|W_{out}\|_F{}^2 \tag{A10}$$

where $\hat{\mathbf{u}}_t = \hat{\mathbf{g}}_t \cup \hat{\mathbf{r}}_t$, $\hat{\mathbf{g}}_t = \hat{\mathbf{g}}_t(\mathbf{g}_{t-1}, \mathbf{v}_t, \xi_t)$ is implemented using a continuous time RNN, with initial state $\hat{\mathbf{g}}_0 = 0$, and subsequent states given by the recurrence relation in *Equation A9* and stationary noise $\xi_t \sim \mathcal{N}(\mu, \sigma^2)$. Moreover, $\hat{\mathbf{r}}_t = W_{out}\hat{\mathbf{g}}_t$, and $W_{in}$ is a weight matrix for the velocity input $\mathbf{v}_t$ to the RNN. The domain $\Omega$ is a 2D square arena visited along simulated trajectories, and the network only received velocity inputs along trajectories, necessitating path integration. In this case, the target representation is Cartesian coordinates $\mathbf{u}(\mathbf{r}_t) = \mathbf{r}_t \in \Omega$. The authors report that the learned recurrent representations $\hat{\mathbf{g}}$ appear square grid, band, and border cell-like.

*Banino et al., 2018* considered the case of a recurrent long short-term memory (LSTM) network trained to do supervised position prediction. Unlike *Cueva and Wei, 2018*, the training objective featured two target representations, $\mathbf{u}_t = \mathbf{p}_t \cup \mathbf{z}_t$. The first target representation, $\mathbf{p}_t = \mathbf{p}(\mathbf{r}_t)$, was given by an ensemble of normalized, Gaussian place-like units, with $\mathbf{r}_t \in \Omega$ being Cartesian coordinates along discretized spatial trajectories in a square domain $\Omega$. The second target representation consisted of an ensemble of units encoding heading direction, $\mathbf{z}_t = \mathbf{z}(\phi_t)$, where $\phi_t$ is the head direction at time $t$. The representations of the head direction ensemble were given by a normalized mixture of von Mises distributions. At each step of path integration, the network received linear and angular velocity information along simulated trajectories. In summary, the loss and corresponding soft constraints can be written as

$$\mathcal{L} = \text{CE}\left(\mathbf{p}_t \,\|\, \hat{\mathbf{p}}_t\right) + \text{CE}\left(\mathbf{z}_t \,\|\, \hat{\mathbf{z}}_t\right),$$
$$\mathcal{C}_1 : \lambda\|W\|_F{}^2 \tag{A11}$$
$$\mathcal{C}_2 : \text{Dropout}(\hat{\mathbf{u}}, \text{rate} = 0.5)$$

where CE is the cross entropy and Dropout *Srivastava et al., 2014* is a method that ablates random units with a specified rate during training to promote redundancy.

The cognitive map, in this case, is given by $\hat{\mathbf{u}}_t = \hat{\mathbf{h}}_t \cup \hat{\mathbf{g}}_t \cup \hat{\mathbf{p}}_t \cup \hat{\mathbf{z}}_t$, with $\hat{\mathbf{h}}_t$ defined by a recurrent neural network with a $\tanh$ activation function, while $\hat{\mathbf{g}}_t = W_g\hat{\mathbf{h}}_t$ is an intermediate linear layer. Finally,

$\hat{\mathbf{p}} = W_p\hat{\mathbf{g}}$, and $\hat{\mathbf{z}} = W_z\hat{\mathbf{g}}$. The intermediate representations, a subset of the cognitive map, $\hat{\mathbf{g}}_t$ were found to display heterogeneous, grid-like responses.

Sorscher et al. reproduced *Banino et al., 2018*; *Cueva and Wei, 2018*; *Dordek et al., 2016* and refined the grid cell model in *Banino et al., 2018* by considering a simpler RNN structure (a vanilla RNN - although other variations have also been tested and shown to provide similar results *Nayebi et al., 2021*), removing head-direction inputs and outputs, the intermediate linear layer, dropout, refining the place cell target representation, and selecting the ReLU as the recurrent activation function *Sorscher et al., 2023*.

$$\mathcal{L} = \mathrm{CE}\left(\mathbf{p}_t \,\|\, \hat{\mathbf{p}}_t\right),$$
$$\mathcal{C}_1 : \lambda\|W\|_2 \tag{A12}$$

where CE is again the cross entropy, $\mathbf{p}_t = \mathbf{p}(\mathbf{r}_t)$ is a difference of softmax place cell encoding of the current position of the virtual agent. In this case, the cognitive map is given by $\hat{\mathbf{u}}_t = \hat{\mathbf{p}}_t \cup \hat{\mathbf{g}}_t$, where $\hat{\mathbf{g}}_t$ and $\hat{\mathbf{p}}_t = W\hat{\mathbf{g}}_t$, where $W$ is a weight matrix. Notably, $\hat{\mathbf{g}}_t$ is computed using a vanilla RNN that learns implicit path integration using Cartesian velocity inputs. The authors report that the recurrent responses $\hat{\mathbf{g}}_t$ learn to exhibit striking hexagonal firing fields, similar to *Dordek et al., 2016*.

This brief taxonomy of normative navigation models hopefully shows how our definition of a cognitive map can be used describe a range of different models that learn biologically inspired representations through the lens of machine learning. Furthermore, our definition, and the notion of a target representation, could hopefully inspire new models. For example, one could consider decoding into a target representation of simulated grid cells.

