## [Editor Report · eLife Assessment]

This **useful** modeling study shows how spatial representations similar to experiment emerge in a recurrent neural network trained on a navigation task by requiring path integration and decodability, but without relying on grid cells. The network modeling results are **solid**, although the link to experimental data may benefit from further development.

---

## [Referee Report · Reviewer #1 (Public review)]

Summary:

This work studies representations in a network with one recurrent layer and one output layer that needs to path-integrate so that its position can be accurately decoded from its output. To formalise this problem, the authors define a cost function consisting of the decoding error and a regularisation term. They specify a decoding procedure that, at a given time, averages the output unit center locations, weighted by the activity of the unit at that time. The network is initialised without position information, and only receives a velocity signal (and a context signal to index the environment) at each timestep, so to achieve low decoding error it needs to infer its position and keep it updated with respect to its velocity by path integration.

The authors take the trained network and let it explore a series of environments with different geometries while collecting unit activities to probe learned representations. They find localised responses in the output units (resembling place fields) and border responses in the recurrent units. Across environments, the output units show global remapping and the recurrent units show rate remapping. Stretching the environment generally produces stretched responses in output and recurrent units. Ratemaps remain stable within environments and stabilise after noise injection. Low-dimensional projections of the recurrent population activity forms environment-specific clusters that reflect the environment's geometry, which suggests independent rather than generalised representations. Finally, the authors discover that the centers of the output unit ratemaps cluster together on a triangular lattice (like the receptive fields of a single grid cell), and find significant clustering of place cell centers in empirical data as well.

The model setup and simulations are clearly described, and are an interesting exploration of the consequences of a particular set of training requirements - here: path integration and decodability. But it is not obvious to what extent the modelling choices are a realistic reflection of how the brain solves navigation. Therefore, it is not clear whether the results generalize beyond the specifics of the setup here.

Strengths:

The authors introduce a very minimal set of model requirements, assumptions, and constraints. In that sense, the model can function as a useful 'baseline', that shows how spatial representations and remapping properties can emerge from the requirement of path integration and decodability alone. Moreover, the authors use the same formalism to relate their setup to existing spatial navigation models, which is informative.

The global remapping that the authors show is convincing and well-supported by their analyses. The geometric manipulations and the resulting stretching of place responses, without additional training, are interesting. They seem to suggest that the recurrent network may scale the velocity input by the environment dimensions so that the exact same path integrator-output mappings remain valid (but maybe there are other mechanisms too that achieve the same).

The simulations and analyses in the appendices serve as insightful controls for the main results.

The clustering of place cell peaks on a triangular lattice is intriguing, given there is no grid cell input. It could have something to do with the fact that a triangular lattice provides optimal coverage of 2d space? The included comparison with empirical data is valuable as a first exploration, showing a promising example, but doesn't robustly support the modelling results.

---

## [Referee Report · Reviewer #2 (Public review)]

Summary:

The authors proposed a neural network model to explore the spatial representations of the hippocampal CA1 and entorhinal cortex (EC) and the remapping of these representations when multiple environments are learned. The model consists of a recurrent network and output units (a decoder) mimicking the EC and CA1, respectively. The major results of this study are: the EC network generates cells with their receptive fields tuned to a border of the arena; the decoder develops neuron clusters arranged in a hexagonal lattice. Thus, the model accounts for entrohinal border cells and CA1 place cells. It suggests that the remapping of place cells occurs between different environments through state transitions corresponding to unstable dynamical modes in the recurrent network.

Strengths:

The authors found a spatial arrangement of receptive fields similar to their model's prediction in experimental data recorded from CA1. Thus, the model proposes plausible mechanisms to generate hippocampal spatial representations without relying on grid cells. The model also suggests an interesting possibility that path integration is not the speciality of grid cells.

Weaknesses:

The role of grid cells in the proposed view, i.e., the boundary-to-place-to-grid model, remains elusive. The model can generate place cells without generating entorhinal grid cells. Moreover, the model can generate hexagonal grid patterns of place cells in a large arena. Whether and how the proposed model is integrated into the entire picture of the hippocampal-entorhinal meｍory processing remains elusive.

---

## [Referee Report · Reviewer #3 (Public review)]

Summary:

The authors used recurrent neural network modelling of spatial navigation tasks to investigate border and place cell behaviour during remapping phenomena.

Strengths:

The neural network training seemed for the most part (see comments later) well-performed, and the analyses used to make the points were thorough.

The paper and ideas were well-explained.

Figure 4 contained some interesting and strong evidence for map-like generalisation as environmental geometry was warped.

Figure 7 was striking and potentially very interesting.

It was impressive that the RNN path-integration error stayed low for so long (Fig A1), given that normally networks that only work with dead-reckoning have errors that compound. I would have loved to know how the network was doing this, given that borders did not provide sensory input to the network. I could not think of many other plausible explanations... It would be even more impressive if it was preserved when the network was slightly noisy.

Update:

The analysis of how the RNN remapped, using a context signal to switch between largely independent maps, and the examination of the border like tuning in the recurrent units of the RNN, were both thorough and interesting. Further, in the updated response I appreciated the additional appendix E which helped substantiate the claim that the RNN neurons were border cells.

---

## [Author Response]

The following is the authors’ response to the previous reviews.

**Reviewer #1:**
In the future, could you please include the exact changes made to the manuscript in the relevant section of the rebuttal, so it's clear which changes addressed the comment? That would make it easier to see what you refer to exactly - currently I have to guess which manuscript changes implement e.g. "We have tried to make these points more evident".

Yes, we apologize for the inconvenience.

On possible navigation solutions:I'm not sure if I follow this argument. If the networks uses a shifted allocentric representation centred on its initial state, it couldn't consistently decode the position from different starting positions within the same environment (I don't think egocentric is the right term here - egocentric generally refers to representations relative to the animal's own direction like "to the left" rather than "to the west" but these would not work in the allocentric decoding scheme here). In other words: If I path integrate my location relative to my starting location s1 in environment 1 and learn how to decode that representation to an environment location, I cannot use the same representation when I start from s2 in environment 1, because everything will have shifted. I still believe using boundaries is the only solution to infer the absolute location for the agent here (because that's the only information that it gets), and that's the reason for finding boundary representations (and not grid cells). Imagine doing this task on a perfect torus where there are no boundaries: it would be impossible to ever find out at what 'absolute' location you are in the environment. I have therefore not updated this part of my review, but do let me know if I misunderstood.

Thank you for addressing this point, which is a somewhat unusual feature of our network: We believe the point you raise applies if the decoding were fixed. However, in our case, the decoding is dynamic and depends on the firing pattern, as place unit centers are decoded on a per-trajectory basis. Thus, a new place-like basis may be formed for each trajectory (and in each environment). Hence, the model is not constrained to reuse its representation across trajectories or environments, as place centers are inferred based on unit firing. However, we do observe that the network learns to use a fixed place field placement in each geometry, which likely reflects some optimal solution to the decoding problem. This might also help to explain the hexagonal arrangement of learned field centers. Finally, we agree that egocentric may not be entirely accurate, but we found it to be the best word to distinguish from the allocentric-type navigation adopted by the network.

Regarding noise injection:Beyond that noise level, the network might return to high correlations, but that must be due to the boundary interactions - very much like what happens at the very beginning of entering an environment: the network has learned to use the boundary to figure out where it is from an uninformative initial hidden state. But I don't think this is currently reflected well in the main text. That still reads "Thus, even though the network was trained without noise, it appears robust even to large perturbations. This suggests that the learned solutions form an approximate attractor." I think your new (very useful!) velocity ablations show that only small noise is compensated for by attractor dynamics, and larger noise injections are error corrected through boundary interactions. I've added this to the new review.

Thank you for your kind feedback: We have changed the phrasing in the text to say “robust even to moderate perturbations. ” As we hold that, while numerically small, the amount of injected noise is rather large when compared to the magnitude of activities in the network (see Fig. A5d); the largest maximal rate is around 0.1, which is similar to the noise level at which output representations fail to re-converge. However, some moderation is appropriate, we agree.

On contexts being attractive:In the new bit of text, I'm not sure why "each environment appears to correspond to distinct attractive states (as evidenced by the global-type remapping behavior)", i.e. why global-type remapping is evidence for attractive states. Again, to me global-type remapping is evidence that contexts occupy different parts of activity space, but not that they are attractive. I like the new analysis in Appendix F, as it demonstrates that the context signal determines which region of activity space is selected (as opposed to the boundary information!). If I'm not mistaken, we know three things: 1. Different contexts exist in different parts of representation space, 2. Representations are attractive for small amounts of noise, 3. The context signal determines which point in representation space is selected (thanks to the new analysis in Appendix F). That seems to be in line with what the paper claims (I think "contexts are attractive" has been removed?) so I've updated the review.

It seems to us that we are in agreement on this point; our aim is simply to point out that a particular context signal appears to correspond to a particular (discrete) attractor state (i.e., occupying a distinct part of representation space, as you state), it just seems we use slightly different language, but to avoid confusion, we changed this to say that “representations are attractive”.

Thanks again for engaging with us, this discussion has been very helpful in improving the paper.

**Reviewer #2:**
However, I still struggle to understand the entire picture of the boundary-to-place-to-grid model. After all, what is the role of grid cells in the proposed view? Are they just redundant representations of the space? I encourage the authors to clarify these points in the last two paragraphs on pages 17-18 of the discussion.

Thank you for your feedback. While we have discussed the possible role of a grid code to some extent, we agree that this point requires clarification. We have therefore added to the discussion on the role of grid cells, which now reads “While the lack of grid cells in this model is interesting, it does not disqualify grid cells from serving as a neural substrate for path integration. Rather, it suggests that path integration may also be performed by other, non-grid spatial cells, and/or that grid cells may serve additional computational purposes. If grid cells are involved during path integration, our findings indicate that additional tasks and constraints are necessary for learning such representations. This possibility has been explored in recent normative models, in which several constraints have been proposed for learning grid-like solutions. Examples include constraints concerning population vector magnitude, conformal isometry (xu, 2022, schaeffer, 2023, schoyen,2024), capacity, spatial separation and path invariance (schaeffer, 2023). Another possibility is that grid cells are geared more towards other cognitive tasks, such as providing a neural metric for space (ginosar, 2023, pettersen, 2024), or supporting memory and inference-making (whittington, 2020). That our model performs path integration without grid cells, and that a myriad of independent constraints are sufficient for grid-like units to emerge in other models, presents strong computational evidence that grid cells are not solely defined by path integration, and that path integration is not only reserved for grid cells.”

Thank you again for your time and input.